# Glyphosate is a transformation product of a widely used aminopolyphosphonate complexing agent

Anna M. Röhnelt [1], Philipp R. Martin [1,5] ✉, Mathis Athmer [2], Sarah Bieger [3], Daniel Buchner [1], Uwe Karst [2], Carolin Huhn [3], Torsten C. Schmidt [4] & Stefan B. Haderlein [1] ✉

Diethylenetriamine penta(methylenephosphonate) (DTPMP) and related aminopolyphosphonates (APPs) are widely used as chelating agents in household and industrial applications. Recent studies have linked APP emissions to elevated levels of the herbicide glyphosate in European surface waters. However, the transformation processes and products of APPs in the environment are largely unknown. We show that glyphosate is formed from DTPMP by reaction with manganese at near neutral pH in pure water and in wastewater. Dissolved $Mn^{2+}$ and $O_2$ or suspended $MnO_2$ lead to the formation of glyphosate, which remains stable after complete DTPMP conversion. Glyphosate yields vary with the reaction conditions and reach up to 0.42 mol%. The ubiquitous presence of manganese in natural waters and wastewater systems underscores the potential importance of Mn-driven DTPMP transformation as a previously overlooked source of glyphosate in aquatic systems. These findings challenge the current paradigm of herbicide application as the sole source of glyphosate contamination and necessitate a reevaluation of water resource protection strategies.

The aminomonophosphonate glyphosate is the most widely used herbicide worldwide[1]. It is a non-selective herbicide, mostly used to kill weeds that compete with crops. Its utilization has significantly increased following the introduction of genetically modified herbicide-tolerant crops, which allowed for extended glyphosate application periods[1,2]. According to the US National Pesticide Information Center (NPIC) the typical half-life of glyphosate is relatively short (1.5 months)[3] but highly variable in soils ranging from few days to several years[4–7]. Environmental persistence of glyphosate is reflected by its frequent detection in ground and surface waters[8–13]. The major transformation product (TP) of glyphosate in the environment is aminomethylphosphonate (AMPA), which exhibits a longer half-life than glyphosate[7,11].

Glyphosate pollution of water bodies so far has exclusively been attributed to herbicide applications[14]. Yet, a recent study revealed negative removal rates for glyphosate and AMPA in municipal wastewater treatment plants[15], while another study highlighted municipal wastewaters as a significant source for glyphosate and AMPA in surface waters in Europe[12]. As the high and rather constant loads of glyphosate and AMPA in wastewater effluents over the year are not compatible with herbicide applications, a different source was suspected[12]. Aminopolyphosphonates (APPs), which are widely used in laundry detergents in the EU[16], are known precursors of AMPA[17–21]. Since the basic structure of glyphosate is already present in certain APPs (see Fig. 1), they are suspected precursors for glyphosate, too[12,22].

[1]Geo- and Environmental Research Center, Department of Geosciences, University of Tübingen, Tübingen, Germany. [2]Institute of Inorganic and Analytical Chemistry, University of Münster, Münster, Germany. [3]Institute of Physical and Theoretical Chemistry, Department of Chemistry, University of Tübingen, Tübingen, Germany. [4]Instrumental Analytical Chemistry and Center for Water and Environmental Research (ZWU), University of Duisburg-Essen, Essen, Germany. [5]Present address: Division of Environmental Geosciences, Centre for Microbiology and Environmental Systems Science, University of Vienna, Vienna, Austria. ✉e-mail: philipp.martin@univie.ac.at; stefan.haderlein@uni-tuebingen.de

Phosphonate consumption (APPs as well as N-free analogues) in Europe was 49,000 metric tons per year in 2012[16]. Detergents and bleaches are major applications for phosphonates. 7613 metric tons were used in household detergent and cleaning applications in Germany in 2019[23]. The three commercially most relevant APPs are aminotris(methylene phosphonate) (ATMP), ethylenediaminetetra(methylene phosphonate) (EDTMP) and diethylenetriamine penta(methylene phosphonate) (DTPMP)[16].

While the predominant removal process for DTPMP and other polyphosphonates in wastewater treatment plants (WWTPs) is commonly attributed to sorption onto sewage sludge[16,24,25], recent studies underscore the need to critically examine also the transformation pathways of these compounds[12,15].

Transformation of APPs (photolysis, Mn²⁺/O₂, MnOOH) with AMPA, iminodi(methylene phosphonate) (IDMP) and phosphate as major TPs (see Fig. 1) is well documented in the literature[17–19,26–28]. However, evidence for glyphosate formation is limited to ozonation of EDTMP in drinking water[22] and is lacking under environmentally relevant conditions. The formation of glyphosate from APPs requires the presence of an ethylene moiety on the nitrogen atom of the APP molecule, whose terminal carbon can ultimately be oxidized to a carboxylic acid. Among the high-volume APPs, only EDTMP and DTPMP exhibit this structural feature, the latter one being quantitatively the most significant[29].

Klinger et al.[22] proposed a reaction scheme for the formation of glyphosate from EDTMP by ozone, a strong oxidant widely used in water treatment. The authors proposed an aldehyde as an intermediate after a C–N bond cleavage within the ethylene moiety and further oxidation of this aldehyde to the carboxylic acid required for the formation of glyphosate[20].

Besides ozone, manganese oxide minerals (MnOₓ) are strong oxidizing agents that are important not only in technical systems but also in the environment[30]. They are often formed by microbial oxidation of dissolved Mn^{II}, leading to amorphous structures with high surface areas[30,31]. Manganese minerals are ubiquitous, not only present in soils and sediments but also in substantial concentrations in sewage sludge[32]. The significance of manganese for the oxidation of ATMP and other APPs including DTPMP has early been recognised by Nowack & Stone[27,28,33]. Recently, the transformation of IDMP by manganese dioxide was investigated in detail revealing AMPA and PO₄³⁻ as main TPs[26]. The transient formation of N-formyl-AMPA (F-AMPA) suggests oxidation of the terminal carbon of an N-methyl-AMPA intermediate to an aldehyde after C–P bond cleavage of IDMP. In analogy to the oxidation of EDTMP by ozone, further oxidation of this terminal carbon might lead to the carboxylic acid – in this case with a methylene moiety as IDMP is the precursor.

Therefore, it is conceivable that manganese oxide minerals or dissolved manganese may lead to glyphosate formation from APPs bearing an ethylene moiety (see Fig. 1) in technical and environmental systems.

Here, we present the results of systematic laboratory experiments on the formation of glyphosate and AMPA as TPs of DTPMP oxidation by manganese dioxide and dissolved manganese and oxygen. The study was designed to capture relevant environmental conditions regarding pH and the presence or absence of dissolved oxygen.

## Results and Discussion

We report the results of laboratory batch experiments designed to elucidate the effects of MnO₂ mineral concentrations (0.1 g/L vs. 1 g/L), dissolved Mn²⁺ (1 mM) and the presence or absence of O₂ (atmosphere/21 vol.-% O₂ vs N₂ atmosphere) on the transformation of 1 mM DTPMP. The experiments were carried out in purified water (buffered at pH 6) as well as in sterile-filtered wastewater (pH 8) as matrix. The pH values were monitored and are depicted in Fig. S1. Control experiments without manganese oxides or dissolved Mn²⁺ generally showed no reactivity towards DTPMP (see Supplementary Fig. S2).

### DTPMP transformation by manganese at pH 6

In aqueous solution buffered at pH 6, DTPMP was completely transformed by MnO₂ within ≤ 24 h under all conditions studied (see Fig. 2). With 1.0 g/L MnO₂ under oxic conditions (fastest reaction kinetics), complete DTPMP transformation was observed after 20 min, while for 0.1 g/L MnO₂ under anoxic conditions (slowest reaction kinetics) complete DTPMP transformation was observed after 24 h. Under oxic conditions with 1 mM Mn²⁺ the slowest transformation kinetics of DTPMP were observed and complete DTPMP transformation was reached only after >130 h (see Fig. S3).

To evaluate the DTPMP transformation kinetics, pseudo 0ᵗʰ-order rate constants were determined as no higher reaction order

**Fig. 1 | Scheme of DTPMP transformation and identified transformation products.** Schematic representation of the formation of phosphate, AMPA, IDMP and glyphosate (proposed) from DTPMP. Phosphate can form via one C–P bond cleavage (iv), IDMP is formed via one C–N bond cleavage (ii), while AMPA is formed via two C–N bond cleavages (i, ii). We propose one pathway for the formation of glyphosate from DTPMP via two C–N bond cleavages (i, iii) and oxidation of the terminal C first to the aldehyde (v) and then to the carboxylic acid (vi). The symmetry of the DTPMP molecule, which contains five phosphonate groups, allows multiple equivalent bond cleavages to lead to the same resultant product. For clarity, only one representative option for each potential cleavage is illustrated. All compounds are depicted in their fully deprotonated forms.

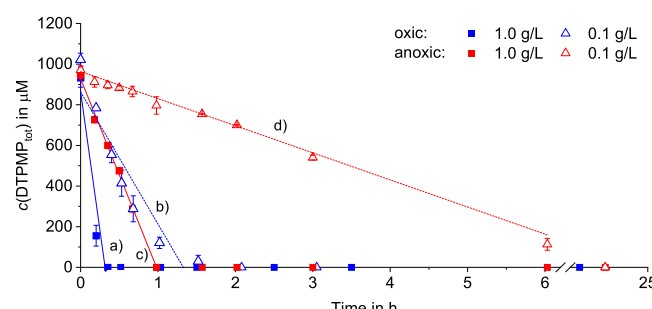

**Fig. 2 | DTPMP transformation by MnO₂ suspended in pure water buffered at pH 6.** Total (aqueous and sorbed) DTPMP concentrations quantified using IC-IPAD as a function of time and pseudo-0ᵗʰ order fits for all four experiments with MnO₂ in aqueous MES buffer (pH 6). **a** 1.0 g/L MnO₂ oxic conditions, (**b**) 0.1 g/L MnO₂ oxic conditions, (**c**) 1.0 g/L MnO₂ anoxic conditions, (**d**) 0.1 g/L MnO₂ anoxic conditions. Error bars represent absolute errors between experimental duplicates.

**Table 1 | Reaction rate constants of DTPMP formation by MnO$_2$ in pure water at pH 6**

| c(MnO$_2$) | 1.0 g/L | 0.1 g/L | 1.0 g/L | 0.1 g/L |
|---|---|---|---|---|
| O$_2$ | oxic | oxic | anoxic | anoxic |
| $k$ in μM/h | 2729 ± 800 | 650 ± 106 | 946 ± 26 | 133 ± 3 |
| $k_{norm}$ in μmol/ (m$^2$ × h) | 42 ± 12 | 101 ± 16 | 14.7 ± 0.4 | 20.6 ± 0.5 |
| R$^2$ | 0.842 | 0.859 | 0.996 | 0.988 |
| Linear section in h | 0–0.35 | 0–1.5 | 0–0.67 | 0–6.0 |

Pseudo-0$^{th}$ order reaction rate constants ($k$) and those reaction rate constants normalized to the surface area ($k_{norm}$) for DTPMP transformation in the four experiments with MnO$_2$ in MES buffer. The standard errors of the linear regression are given as ±x. R$^2$ is the regression coefficient of the linear regression from the start of the experiment until complete DTPMP transformation (24 h excluded for 0.1 g/L anoxic due to low data point density). The linear section indicates the time interval/timepoints included in the linear regression.

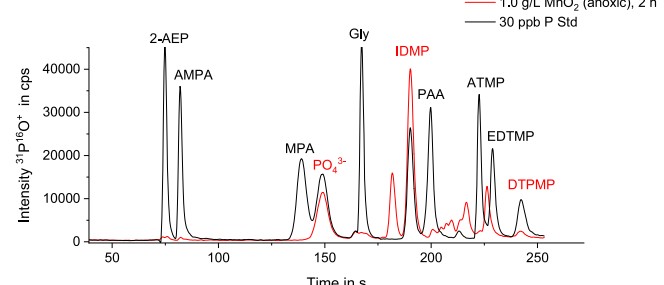

**Fig. 3 | Chromatogram of DTPMP transformation products.** Phosphorus-selective IC-ICP-MS chromatogram of an exemplary sample overlayed by the chromatogram of a standard mix including the denoted compounds. The sample (aqueous fraction of duplicate A of the experiment containing 1.0 g/L MnO$_2$ under anoxic conditions, reaction time 2 h) is presented in red color, while the standard mix (30 ppb P per compound) is presented in black. The sample was diluted 1:1000 to match the calibration range. Abbreviations for standard compounds not described in the text: 2-AEP = 2-aminoethylphosphonate, MPA methylphosphonate, PAA phosphonoacetate.

adequately described the kinetics across all four MnO$_2$ experiments. These constants were derived by linear regression considering the time intervals described in Table 1. This approach allowed for a comparative kinetic analysis of the four MnO$_2$ experiments.

In the absence of dissolved oxygen, the transformation of DTPMP followed a pseudo-0$^{th}$ order rate law with R$^2$ values of > 0.98. The presence of dissolved oxygen accelerated the reaction and changed the rate law as indicated by the deviation from pseudo-0$^{th}$ order kinetics (R$^2 \cong$ 0.85). Both MnO$_2$ concentration and the presence of O$_2$ concentrations enhanced DTPMP transformation kinetics, with the fastest rates for 1.0 g/L MnO$_2$ under oxic conditions, followed by 1.0 g/L MnO$_2$ under anoxic conditions. Both experiments (oxic and anoxic) containing 0.1 g/L MnO$_2$ exhibited slower reaction kinetics than 1.0 g/L, while the reaction containing 0.1 g/L under anoxic conditions showed the slowest reaction. Normalization of the reaction rate constants to the specific surface area ($k_{norm}$) provided further insights into the role of oxygen and available surface area. $k_{norm}$ values showed that the oxic experiments were faster compared to their anoxic counterparts.

The rate enhancing role of oxygen may be related to multiple processes involving different redox states of manganese as reported in literature. Under oxic conditions, Mn$^{2+}$ (formed by the reduction of Mn$^{IV}$[26,30]) is known to catalyze DTPMP (and ATMP, EDTMP) oxidation by O$_2$ (Mn$^{2+}$/O$_2$) in homogeneous solution[28,34]. However, the much slower reaction kinetics of DTPMP in homogeneous solution (1 mM Mn$^{2+}$ and O$_2$) compared to the heterogeneous systems found in this study clearly show that the strongly enhancing role of O$_2$ in MnO$_2$ experiments must be due to processes other than mere Mn$^{2+}$/O$_2$ interaction, probably involving the formation of Mn$^{III}$ on the mineral surface.

Manganese minerals with elevated Mn$^{III}$ content appear to be more reactive oxidants[30,35,36]. In heterogeneous systems containing Mn$^{2+}$ and MnO$_2$, Mn$^{III}$ can be formed by comproportionation and be associated with the mineral surface or reside in solution[35]. Furthermore, MnO$_2$ can catalyze the oxidation of Mn$^{II}$ by O$_2$ to Mn$^{III}$[37]. Finally, MnO$_2$ itself may act as a direct oxidant but also as a catalyst in connection with O$_2$[38].

Previous research on IDMP transformation using the same sample of MnO$_2$ demonstrated that roughly two electrons are accepted by MnO$_2$ per transformation of one IDMP molecule[26]. Thus, the electron-accepting capacity of 0.1 g/L MnO$_2$, which corresponds to 1.1 mM MnO$_2$, cannot explain the complete transformation of 1 mM DTPMP to smaller transformation products, which are partially further oxidized. Moreover, as normalized reaction rate constants both under oxic and anoxic conditions were higher for low mineral concentrations (0.1 g/L vs. 1.0 g/L) a catalytic role of MnO$_2$ next to its direct oxidation activity is evident.

## Formation of TPs from DTPMP

During DTPMP transformation, the formation of various phosphorus-containing TPs was monitored in the aqueous phase using ion chromatography (IC) coupled to inductive-coupled plasma mass spectrometry (ICP-MS). Figure 3 shows an exemplary chromatogram (1 g/L MnO$_2$, anoxic conditions, reaction time of 2 h, aqueous phase) next to a mix-standard of 30 ppb P (0.97 μM) per compound. The main TPs identified based on retention times and reference compounds in all experiments were IDMP and phosphate, consistent with previous studies[16–18,28]. Based on the initial DTPMP concentration, IDMP formation reached up to 97 mol-% (0.1 g/L MnO$_2$, anoxic conditions), while PO$_4^{3-}$ formation peaked at 153 mol-% (1.0 g/L MnO$_2$, oxic conditions). Regarding phosphate, the maximum molar yield amounts to 500 mol-%, due to DTPMP's five phosphonate moieties. DTPMP, IDMP, and PO$_4^{3-}$ concentrations over time in the aqueous phase are depicted in Supplementary Fig. S4.

In the exemplary chromatogram shown in Fig. 3, a double peak is visible at the retention time of AMPA (83.0 s), while a triple peak is observed around the retention time of glyphosate (167.5 s). Both peaks, even if considered to be the respective compounds, were below the instrumental LODs (0.9 ppb P for AMPA and 1.7 ppb P for glyphosate) and therefore represent an almost negligible fraction within the TP spectrum. Thus, it is not surprising that the formation of glyphosate has so far been mostly overlooked.

To verify glyphosate and AMPA formation during DTPMP transformation, we used FMOC derivatization and subsequent quantification by means of reversed-phase high-performance liquid chromatography (RP-HPLC) coupled to a triple-quadrupole (QQQ) mass spectrometer, an established trace analysis method for glyphosate and AMPA[39–41].

## Glyphosate and AMPA formation from DTPMP

The formation of glyphosate as well as AMPA was observed in all experiments with MnO$_2$ (see Fig. 4). While AMPA is the main TP of glyphosate in the environment[7,11,42], the main path for AMPA formation from DTPMP is via two C–N bond cleavages (see Fig. 1).

Glyphosate and AMPA yields were calculated in mol-% based on the analyzed initial DTPMP concentration. The maximum molar yields for the experiments conducted at pH 6 within the timespans presented in Fig. 4 were 0.16 mol-% (1.6 μM) for glyphosate and 10.13 mol-% (95 μM) for AMPA. Clearly, the concentration of oxygen and MnO$_2$ affected the glyphosate and AMPA formation kinetics and maximum observable yields.

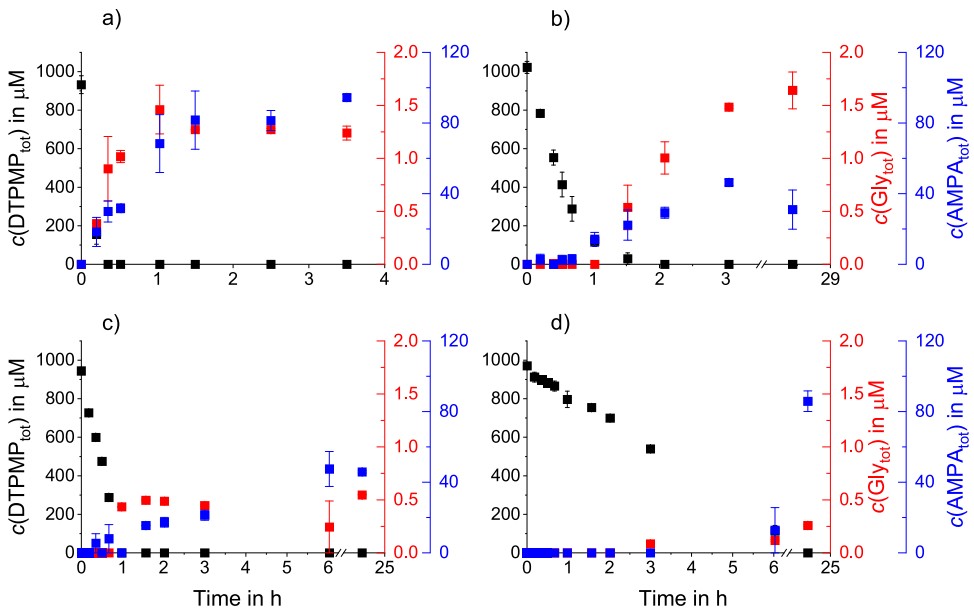

**Fig. 4 | Kinetics of DTPMP transformation and product formation at different reaction conditions.** Total DTPMP, glyphosate and AMPA concentrations during DTPMP oxidation by $MnO_2$ in four different experiments in MES buffer (pH 6). DTPMP (black) was quantified using IC-IPAD (see Fig. 2), AMPA (blue) and glyphosate (red) using LC-QQQ. **a** 1.0 g/L $MnO_2$ oxic, (**b**) 0.1 g/L $MnO_2$ oxic, (**c**) 1.0 g/L $MnO_2$ anoxic, (**d**) 0.1 g/L $MnO_2$ anoxic. Error bars represent absolute errors between experimental duplicates.

**Table 2 | Maximum yields of glyphosate and AMPA at various experimental conditions**

| $c(MnO_2)$ in g/L | $c(Mn^{2+})$ in mM | $O_2$ | Matrix | $AMPA_{max}$ in mol-% | Timepoint in h | $Gly_{max}$ in mol-% | Timepoint in h |
|---|---|---|---|---|---|---|---|
| 1.0 | - | oxic | MES | 10.1 ± 0.2 | 3.5 | 0.16 ± 0.02 | 1.0 |
| 0.1 | - | oxic | MES | 4.5 ± 0.2 | 3.0 | 0.16 ± 0.01 | 28 |
| 1.0 | - | anoxic | MES | 5.0 ± 1.0 | 6.0 | 0.06 ± 0.00 | 24 |
| 0.1 | - | anoxic | MES | 10.1 ± 0.5 | 24 | 0.03 ± 0.00 | 24 |
| 1.0 | - | oxic | WW | 10.3 ± 0.3 | 168 | 0.06 ± 0.01 | 168 |
| - | 1.0 | oxic | MES | 6.7 ± 0.7 | 185 | 0.07 ± 0.00 | 185 |
| - | 1.0 | oxic | WW | 27.1 ± 0.5 | 240 | 0.42 ± 0.01 | 240 |

Maximum total AMPA and glyphosate yields given in mol-% of the initial quantified DTPMP concentration at the denoted timepoints. Errors for AMPA and glyphosate yields represent absolute errors between duplicates. MES denotes experiments in aqueous 20 mM MES buffer at pH 6, while WW stands for sterile-filtered wastewater at pH 8. Timepoint denotes the time of maximum observed AMPA resp. glyphosate formation.

Both compounds formed fastest in the presence of 1.0 g/L $MnO_2$ under oxic conditions (Fig. 4a), reaching their maximum concentration after 1 h. A similar maximum concentration of glyphosate was reached with 0.1 g/L $MnO_2$ under oxic conditions but only after approximately 28 h compared to 1 h (Fig. 4b).

With 0.1 g/L $MnO_2$ under anoxic conditions (Fig. 4d) no AMPA was observed within the first four hours. After 6 h, AMPA was detected in comparably low concentrations around 12 μM increasing up to 98 μM after 24 h, while a maximum glyphosate yield of 0.3 μM was detected after 24 h.

Continued glyphosate and AMPA formation after complete DTPMP consumption (see Fig. 4a–c) suggests the presence of intermediates that are further transformed to glyphosate and/or AMPA. The presence of such intermediates is further supported by a lag phase before glyphosate and AMPA formation, as observed for 0.1 g/L $MnO_2$ under oxic conditions (Fig. 4b). Glyphosate and AMPA concentrations only rose after 1.5 and 1 h, respectively, even though DTPMP was almost completely transformed at that time.

The maximum concentrations of AMPA and glyphosate formed differed between the four experiments (see Table 2). Under anoxic conditions, significantly lower glyphosate yields (0.03 and 0.06 mol-%) compared to oxic conditions (both 0.16 mol-%) were observed, either

due to lower glyphosate formation or rapid subsequent transformation. However, for AMPA this trend is not discernible (see Table 2), potentially due to the numerous reaction pathways leading to AMPA.

To elucidate the significance of heterogeneous ($MnO_2$) and homogeneous ($Mn^{2+}/O_2$) oxidation reactions on product formation, an experiment with 1 mM DTPMP and 1 mM dissolved $Mn^{2+}$ ($MnCl_2$) was conducted under oxic conditions (buffered at pH 6). Neither glyphosate nor AMPA formation was observed within the first 24 h (see Fig. 5a). After 137 h (approximately 5.5 days), however, 6.3 ± 0.2 mol-% AMPA and 0.06 ± 0.01 mol-% glyphosate were quantified. AMPA and glyphosate concentrations stayed almost constant until 185 h yielding 6.8 ± 0.7 mol-% (AMPA) and 0.07 ± 0.00 mol-% (glyphosate).

To account for longer reaction times, the experiment for the most reactive system (1 g/L $MnO_2$, oxic) was repeated but now sampled over 96 h (see Supplementary Fig. S5). AMPA concentrations increased until the end of the experiment and reached a maximum of 206 μM (24 mol-%) after 96 h (4 days). Glyphosate was detected up to a maximum of 1.1 μM (0.1 mol-%), but no clear trend in concentrations could be deduced.

This experiment demonstrates that AMPA and glyphosate – even in the most reactive suspension after 4 days – are not completely transformed. This is remarkable, as the oxidation of glyphosate and

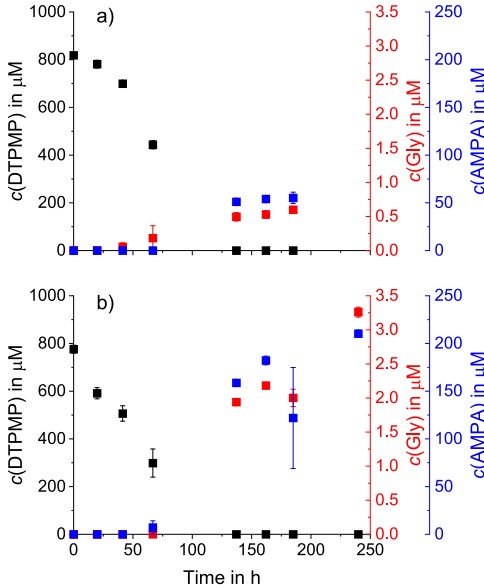

**Fig. 5 | Kinetics of DTPMP transformation and product formation by dissolved Mn²⁺.** DTPMP, glyphosate and AMPA concentrations during oxidation of 1 mM DTPMP by 1 mM Mn²⁺ and oxygen in two different matrices. The matrices consisted of (**a**) ultrapure water with 20 mM MES buffer (pH 6) and (**b**) sterile-filtered wastewater (pH 8). DTPMP (black) was quantified using IC-IPAD, glyphosate (red) and AMPA (blue) were quantified using LC-QQQ. Error bars represent absolute errors between experimental duplicates.

AMPA on manganese oxides has been extensively studied and both compounds can be oxidized by $MnO_x$[43–46]. Thus, further investigations are required to better understand the stability and further transformation of glyphosate and AMPA in consecutive reactions. The accumulation of TPs during the experiment creates a complex matrix that may impede the reaction between AMPA or glyphosate and $MnO_2$. These TPs potentially occupy the mineral surface, reducing the active sites available for further reactions. Two scenarios are consistent with these findings: (i) the built-up of a TP matrix hinders glyphosate and AMPA transformation by diminishing the reactivity of $MnO_2$ through sorption, or (ii) the rate of glyphosate and AMPA formation exceeds their transformation rate in parallel reactions.

### Formation of glyphosate and AMPA from DTPMP in wastewater matrix

To address the environmental relevance of the observations in pure water, experiments containing 1 g/L $MnO_2$ and 1 mM Mn²⁺ were conducted in wastewater (pH 8, sterile filtrated; see the Methods section for details on the wastewater sample). In control experiments with unspiked wastewater with and without $MnO_2$ or Mn²⁺, negligible glyphosate and AMPA concentrations were occasionally detected (see Fig. S6). Dissolved manganese in the wastewater sample was below the detection limit ( < 0.04 mg/L).

DTPMP transformation kinetics with 1.0 g/L $MnO_2$ in the wastewater matrix were slower than in MES buffer at pH 6 (see Fig. S7 a versus b) in line with a lower oxidation potential of manganese oxides at higher pH[47,48], as well as increased electrostatic repulsion between DTPMP and the mineral surface (point of zero charge of 5.6)[30,49]. Furthermore, differences between the experiments with MES buffer and wastewater are likely due to the complex wastewater-matrix containing organics (such as other complexing agents) and cations (e.g., calcium), which were reported to influence APP transformation and sorption[18,50]. Glyphosate and AMPA yields with 1 g/L $MnO_2$ in wastewater were up to 0.06 mol-% (glyphosate) and 10.3 mol-% (AMPA) after 168 h. However, in wastewater spiked with 1 mM Mn²⁺, DTPMP

transformation kinetics were faster and had higher glyphosate and AMPA yields compared to MES-buffered pure water at pH 6 (see Fig. 5.) In wastewater spiked with 1 mM Mn²⁺, the highest glyphosate yield (0.42 mol-%) of all experiments conducted in this study was observed (see Table 2), albeit only after 240 h. Possibly, the reaction kinetics are faster at pH 8 due to stronger complex formation of Mn^II and DTPMP, such as shown for ATMP[28]. At higher pH, less protons compete with metal ions present in solution and DTPMP is more negatively charged[51]. Furthermore, Mn^III complexes might play a role at higher pH. While the stability of Mn^III complexes varies depending on the ligand, certain ligands including desferrioxamine B show higher stability of Mn^III complexes at pH values between 7 and 11[52].

### Environmental implications

Polyphosphonates as replacements for polyphosphates and polycarboxylates in detergents, laundry products and other applications have been considered critical due to their persistence. However, they have been accepted in applications because they were not considered toxicologically relevant. Despite their reportedly high recalcitrance attributed to the high stability of the C–P bond[29,53–55], APPs can be transformed in the presence of manganese, which occurs ubiquitously in WWTPs, soils & sediments, yielding phosphate, IDMP and AMPA as major products.

Our study demonstrates that manganese potentially plays a key role in converting the widely used complexing agent DTPMP in a multi-step reaction to the herbicide glyphosate. The reaction proceeds at circumneutral pH at $MnO_2$ minerals both in the absence and presence of dissolved oxygen but also in homogeneous solution in the presence of Mn²⁺/$O_2$, even in wastewater. Under all conditions studied, AMPA and glyphosate were transformation products, AMPA up to 27.1 mol-% and glyphosate up to 0.42 mol-%. Both the kinetics and yield of DTPMP transformation products were heavily influenced by the reaction conditions ($MnO_2$ concentration, presence of $O_2$ and pH/matrix). Once DTPMP was completely transformed, the concentrations of glyphosate and AMPA remained constant or even increased within 1 to 10 days of kinetic experiments. The persistence of glyphosate and AMPA towards the end of the experiment in the presence of excess $MnO_2$ is remarkable as both compounds can be oxidized by unreacted $MnO_2$ minerals[42–46].

The comparatively high glyphosate yield of the experiment containing 1 mM Mn²⁺ in wastewater under oxic conditions, relative to all other experiments, underlines the relevance of both $MnO_2$ and aqueous Mn²⁺ in environmental and technical systems. This finding underlines the relevance of manganese-driven oxidation reactions in glyphosate and AMPA formation from DTPMP. The distribution between Mn^II and Mn^III/Mn^IV species in the environment is known to depend on various conditions such as pH, microbial activity, and $O_2$ levels[56,57]. Our results demonstrate that both Mn^II/$O_2$ and Mn^IV can lead to glyphosate and AMPA formation, suggesting that these processes may occur under a wide range of environmental conditions.

Overall, this study provides experimental evidence for conversion of a widely used non-toxic commodity compound into a highly debated pesticide[1,2,5,58] under environmentally relevant reaction conditions. While we could demonstrate that the reaction is chemically feasible in the laboratory, future research should elucidate in detail how environmental conditions affect glyphosate formation from DTPMP and related APPs, the formation and identification of key intermediates and field studies including yields in wastewater treatment plants.

In addition, our findings may also provide clues for revisiting APP biotransformation studies. All bacterial growth media used in published APP biotransformation studies contain dissolved manganese[55,59–63]. Thus, manganese-driven oxidation may occur in parallel with or instead of biotransformation of DTPMP and other APPs. Martin et al. (2022)[34] showed that even at a molar ratio of 1:100 (Mn:ATMP), ATMP was completely degraded within 30 h. Thus, it is

conceivable that in biotransformation studies under oxic conditions in the presence of dissolved $Mn^{2+}$, microorganisms may utilize chemical transformation products of APPs, such as phosphate, IDMP, or AMPA, as phosphorus source rather than or in addition to directly metabolizing the target APPs.

Overall, our work offers a scientific basis to rationalize recent and unexpected findings of elevated glyphosate concentrations in European WWTP effluents[12,15] and suggests that manganese may play a crucial role in this phenomenon, potentially serving as a key factor in understanding the underlying processes.

## Methods

### Chemicals

All chemicals were purchased from Merck (Darmstadt, Germany) in the highest available purity, if not described differently. The manganese dioxide (Manganese$^{IV}$oxide, ≥98%; $MnO_2$, Batch No. 168267405) was purchased from Carl Roth (Karlsruhe, Germany). $MnCl_2$ tetrahydrate (p.a., ≥99%) was purchased from Merck (Darmstadt, Germany) The $MnO_2$ specifications can be found in "Methods – Mineral Characterization" and the Supplementary Information. DTPMP was purchased as solid acid from Zschimmer and Schwarz (Lahnstein, Germany) under the name "Cublen D 900 GR" (CAS: 15827-60-8). In order to ascertain the purity of the DTPMP, a $^{31}$P-{$^{1}$H}-NMR measurement was conducted, showing a purity of >98.6% regarding P (nuclear magnetic resonance spectroscopy (NMR) measurements, for details, see below). Glyphosate PESTANAL (≥98.0%), AMPA (99%), IDMP (≥97%), ATMP (≥97.0%), 2-aminoethylphosphonic acid (99 %), methylphosphonic acid (99 %) and phosphonoacetic acid (98 %) solids were purchased from Sigma Aldrich (St Louis, MO, USA). EDTMP was purchased as solid acid from Zschimmer and Schwarz (Lahnstein, Germany) under the name "Cublen ELC 950". The purity of EDTMP was determined to be >98.6 % regarding P using $^{31}$P-{$^{1}$H}-NMR (see below). Isotope-labelled glyphosate (i) and AMPA (ii) were purchased from LGC Standards Ltd. (Teddington, England) (i) and HPC Standards GmbH (Cunnersdorf, Germany) (ii). Phosphate standard solution (1000 mg/L $PO_4^{3-}$ in $H_2O$) for preparation of IC-ICP-MS phosphonate standards was purchased from Merck (Darmstadt, Germany).

Sodium hydroxide for IC-IPAD eluent preparation and analyte desorption from the manganese dioxide was purchased as a 49–51% aqueous solution from Supelco (Merck, Darmstadt, Germany), $NaH_2PO_4$ (p.a., ≥99%) for analyte desorption from Roth (Karlsruhe, Germany) and sodium acetate trihydrate for IC-IPAD eluent preparation from Chemsolute (Renningen, Germany). 2-(N-morpholino)ethanesulfonic acid (MES) buffer (≥99%) for the DTPMP transformation experiments was purchased from Carl Roth. Sodium acetate for IC-IPAD eluent preparation was delivered by Chemsolute (Renningen, Germany).

The cation exchange resin in proton form (Dowex 50 W X 8, 100–200 mesh, ≥1.7 eq/L) used to remove dissolved manganese was purchased from Carl Roth.

The water used for the experiments and IC-IPAD measurements was purified by an ultrapure water purification system (Barnstead, GenPure Pro, Thermo Scientific, Waltham, MA, USA) down to a conductivity below 0.06 μS/cm. For IC-ICP-MS analysis (dilutions and eluent preparation), doubly distilled water from an Aquatron A4000D system (Barloworld Scientific, Nemours, France) was used.

Fluorenylmethoxycarbonyl chloride (FMOC Cl, 98 %) and sodium tetraborate decahydrate (borate, p.a.) used for the derivatization of glyphosate and AMPA were purchased from Carl Roth resp. Honeywell/Fluka (Charlotte, NC, USA). Dichloromethane (DCM, HPLC grade) for washing the derivatized samples was bought from Fisher Scientific (Waltham, MA, USA). For LC analysis with a triple quadrupole mass spectrometer (LC-QQQ) the eluent was prepared using LC/MS-grade acetonitrile (ACN, ≥99.9%) from Honeywell/Riedel-de Haën (Seelze, Germany), LC/MS grade water (Fisher Scientific,

Loughborough, UK) and NH4Ac (p.a., ≥98%) from Sigma Aldrich (St Louis, MO, USA).

Nitric acid ($HNO_3$, 65%, for analysis) for IC-ICP-MS measurements was purchased from Thermo Fisher Scientific (Bremen, Germany) and purified with a DST-1000 acid purification system from Savillex (Eden Prairie, MN, USA). For IC-ICP-MS eluent preparation, aqueous ammonia solution (25–27 %, for trace analysis) was purchased from VWR International LLC (Radnor, PA, USA) and diethylenetriaminepentaacetic acid (DTPA) from Honeywell/Fluka (Charlotte, NC, USA). The IC-ICP-MS post-column internal standard (1000 μg/L indium in 2% aqueous $HNO_3$) was purchased from Sigma-Aldrich (St. Louis, MO, USA).

Deuterium oxide ($D_2O$, 99.9 atom% D) for NMR measurements was obtained from Sigma- Aldrich (Steinheim, Germany).

### Design of DTPMP transformation experiments

The experiments (duplicates with one control) were conducted in 50 mL centrifugation tubes (polypropylene, Fisher Scientific, Waltham, MA, USA) in the presence of ambient air or in the glovebox ($N_2$ atmosphere, $c(O_2) < 10$ ppm) (Unilab from MBRAUN, Garching, Germany) at room temperature ($21 \pm 1$ °C). For the glovebox experiments, DTPMP stock solution, MES buffer solution and deionized water were purged with $N_2$ for one hour under rapid stirring before transfer into the glovebox. Then, DTPMP, MES buffer and water were mixed to yield concentrations of 1 mM DTPMP and 20 mM MES. After mixing, the first aliquot of 5 mL was taken as $t_0$ sample. Then, solid $MnO_2$ was added, to reach a concentration of 0.1 or 1.0 g/L. The reaction suspensions were shaken in an overhead-shaker at a speed of 25 rpm. At defined time points derived from pilot tests, a well suspended 4 mL aliquot of the suspension was taken, centrifuged (15 min, 20,000 rcf), and filtered (0.2 μM PES syringe filter, BGB Analytik, Lörrach, Germany)[26]. To desorb residual analytes from the mineral pellet after centrifugation and sampling the supernatant (aqueous phase), the mineral pellet was treated with 0.1 M NaOH and 0.1 M $NaH_2PO_4$ in the ultrasonic bath for 30 minutes[64]. To account for possible parallel homogenous $Mn^{II}$-catalyzed oxidation by $O_2$ initiated by formed $Mn^{2+}$, an additional experiment with 1 mM $MnCl_2$ (no $MnO_2$) under oxic conditions was conducted. The experiments using 1.0 g/L $MnO_2$ and 1 mM $Mn^{2+}$/$O_2$ as oxidant/catalyst were repeated in wastewater (pH 8).

The homogeneous reactions were quenched by the addition of cation exchange resin to the aliquot taken from the reaction solution to bind $Mn^{2+}$.

Samples were stored in the dark at −20 °C until analysis. Prior to analysis, samples were thawed, treated with cation exchange resin (if not done before freezing) and diluted for the respective measurement purpose.

The wastewater was sampled at 10 am on September 9, 2024, from the municipal wastewater treatment plant in Lustnau (Tübingen, SW Germany), after the screen and grit chamber, but before the primary settling tank. The wastewater was then filtered with different filter systems: I) coffee filter (Melitta, Minden, Germany), II) folded filters 595 ½ (Whatman Int. Ltd, Buckinghamshire, UK), III) glass fibre round filters GF 55 (Schleicher & Schuell, Dassel, Germany) and finally IV) sterile S-PAK 0.22 μm filters (Merck, Darmstadt, Germany). This filtered wastewater was used undiluted as matrix for the experiments. The initial pH of the wastewater was 8. The changes in pH development over time are shown in Fig. S1. Detailed information regarding the composition of the wastewater is provided below.

### Quantification of DTPMP

Quantification of DTPMP was performed according to the IC-IPAD (ion chromatography coupled to integrated amperometric detection) method published by Röhnelt et al. (2025)[65]. A 930 Compact IC Flex ion chromatograph (Metrohm, Herisau, Switzerland) was used, equipped

with an anion exchange column (Dionex IonPac AS16, $2 \times 250$ mm, Thermo Fisher Scientific, Waltham, United States), a suitable guard column (Dionex IonPac AG16, $2 \times 50$ mm) and an amperometric detector (Wall-Jet Cell, Metrohm). The Wall-Jet cell was equipped with a gold working electrode, a Pt auxiliary electrode and an Ag/AgCl reference electrode (all Metrohm). The detector method potential vs. time profile is depicted in Supplementary Fig. S8. The dosing units for i) sample uptake and ii) concentration gradient were both of the type "800 Dosino" (Metrohm), with i) 2 mL and ii) 5 mL cylinder volume.

To prevent $CO_2$ dissolution into the eluents, an overpressure of 0.5 bar $N_2$ was applied to both eluent bottles (gas-tight plastic bottles, Metrohm). 15 mM NaOH served as eluent A, while 50 mM NaOH with 400 mM sodium acetate served as eluent B. The gradient profile is depicted in Supplementary Table S1.

Quantification was performed by external calibration (0.01–20 µM) and normalization to repeatedly injected control standards.

Total DTPMP concentrations were analyzed by combining sorbed and aqueous fractions prior to measuring.

### Quantification of AMPA and glyphosate

AMPA and glyphosate were quantified using reversed phase (RP) liquid chromatography (LC) coupled to triple quadrupole mass spectrometry (QQQ-MS) after derivatization using FMOC chloride[39–41]. The sample was diluted 1:2 (glyphosate) respectively 1:100 (AMPA) with ultrapure water and the respective isotope-labelled standard was added to reach a final concentration of 50 µg/L. Then, 10–15 mg of the cation exchange resin were added to 1 mL of the diluted sample and shaken for 30 min. After sedimentation of the resin, 900 µL of the supernatant were sampled and mixed with 200 µL 80 mM borate buffer and 200 µL 40 mM FMOC chloride in acetonitrile. The sample, which turned milky immediately, was let to rest for 30 min. Afterwards, the clear aqueous phase was washed twice using 2 mL DCM, each. The now derivatized samples were stored in the dark at 4 °C until measurement. Repeated measurements showed the stability of the derivatized samples over several months.

The liquid chromatography (1290 Infinity II, Agilent, Santa Clara, CA, USA) was coupled to a triple quadrupole (QQQ) mass spectrometer (6470 LC/TQ, Agilent, Stadt/LAND), equipped with an AJS source. A reversed phase column (Agilent Poroshell 120 EC-C18 2.7 µm, 2.1 x 100 mm + 2.1 x 5 mm pre-column) was used to separate the derivatized compounds. The derivatized analytes were measured in positive ionization mode with a cell accelerator voltage of 5 V.

2.5 mM aqueous ammonium acetate (A) and acetonitrile (B) served as mobile phases. The concentration gradient profile is provided in Supplementary Table S2. A sample of 1 µL was injected together with 0.2 µL internal standard (200 µg/L glufosinate-FMOC). The column was heated to 40 °C and the flow rate was set to 0.3 mL/min. The MS parameters and fragment ions used for quantification can be found in Supplementary Table S3.

Limits of detection (LOD) and quantification (LOQ) were calculated for each measurement sequence based on the standard deviation of the linear response and the slope of the calibration curve (5–200 µg/L (glyphosate), 5–500 µg/L (AMPA), ISTD 50 µg/L each) following the International Council for Harmonisation (ICH) Q2(R1) guideline[66]. Table S4 provides the LOD/LOQ values derived for every measurement sequence.

The aqueous and sorbed fractions were analyzed separately, with the sorbed fractions representing a minor part of the investigated compounds (see Supplementary Fig. S9).

### Quantification of phosphorus-containing TPs using IC-ICP-MS

For the detection of all phosphorus-containing compounds, a prep-FAST IC system (Elemental Scientific, Omaha, NE, USA) was connected to an iCAP TQ inductively coupled plasma mass spectrometer (ICP-MS) (Thermo Fisher Scientific, Bremen, Germany).

The prepFAST IC system was equipped with an anion exchange column CF-Cr-01 ($50 \times 4$ mm) (Elemental Scientific, Omaha, NE, USA) and the injection volume was set to 50 µL. The chromatographic separation was performed with a flow rate of 1000 µL/min and a post-column internal standard flow rate of 100 µL/min indium in 2% nitric acid (1 µg/L). 300 µg/L DTPA in water (pH 9.2) served as eluent A, while eluent B consisted of 150 mM ammonium nitrate with 300 µg/L DTPA in water (pH 9.2). For the chromatographic separation, a five-step gradient was employed, which is described in Table S5.

Analysis was conducted in QQQ mode with oxygen as a reaction gas. Phosphorus was detected as $^{31}P^{16}O^+$ and indium as $^{115}In^+$ with individual dwell times of 100 ms. Quantification was performed by external calibration. Unknown TPs were quantified using the calibration function of the nearest polyphosphonate standard. The LOD was determined by the $3\sigma$ criteria, and the LOQ was determined by the $10\sigma$ criteria respectively.

Due to the desorption protocol using 0.1 M $NaH_2PO_4$ as competitive sorbent, with this method, just the aqueous phase of the heterogeneous experiments could be analyzed.

### Nuclear Magnetic Resonance Spectroscopy (NMR)

To assure the purity of the DTPMP (EDTMP) purchased commercially, a $^{31}P$-{$^1H$}-NMR measurement was conducted (NMR department, Chemistry Department, University of Tübingen). $^{31}P$-NMR-spectroscopy is a suitable analytical tool to characterize the purity of phosphonates, due to the 100% natural abundance of $^{31}P$ and the wide range of chemical shifts and high sensitivity of the method[67].

10 mg DTPMP (EDTMP) and 600 µL of deuterated water ($D_2O$) were mixed and vortexed for 5 s. Afterwards, 600 µL were transferred to an NMR glass tube. The measurement was performed on a Bruker AMX 600 MHz NMR spectrometer (Bruker, Billerica, MA, USA), operating at 242.94 MHz for phosphorous observation with a zgpg30 pulse program. The acquisition parameters used for this experiment with 1D sequence with power-gated decoupling and a 30 ° flip angle were as follows: number of scans: 64, spectral width: 96153.84 Hz, offset (O1): −12146.85 Hz, acquisition time: 0.34 seconds, relaxation delay (d1): 2.00 s. The spectrum was quantitatively evaluated using the Bruker Top Spin 4.1.4 software.

### DTPMP

In the $^{31}P$-{$^1H$}-NMR-spectrum (Fig. S10), two main signals with an intensity ratio of 1:4 can be seen, which represent the phosphonate-group in the middle of DTPMP (δ (ppm): 12.94) and the four phosphonate groups of DTPMP attached to the outer amine-moieties (δ (ppm): 9.23). Impurities are marked with an asterisk. The sum of all signal-integrals is normalized to 100. Impurities of DTPMP contributed with 1.37%. Thus, the purity of the DTPMP regarding the P content was > 98.6%.

### EDTMP

In the $^{31}P$-{$^1H$}-NMR-spectrum of EDTMP, shown in Fig. S11, we can see one signal attributed to EDTMP representing all chemically equivalent phosphonate-groups (δ (ppm): 8.76). Impurities are marked with an asterisk. The sum of all signal-integrals is normalized to 100. Impurities containing phosphorous of the analyzed EDTMP amount to only 3.40%. The used EDTMP in the experiments is therefore of high purity with 96.6%.

### Mineral characterization

The point of zero charge ($pH_{PZC}$) of the manganese dioxide was determined to be pH $5.6 \pm 0.1$ by zeta potential measurements.

The X-ray diffractogram (see Fig. S12) showed a mostly amorphous structure, interspersed with some crystalline domains (pyrolusite, akhtensite). The specific surface area (SSA) was determined to be $64.5 \pm 0.2 \, m^2/g$ using the Brunauer-Emmett-Teller (BET) method. Measurement details as well as further specifications can be found in the Supplementary Information.

## Wastewater characterization

The wastewater sample taken from the WWTP in Lustnau (Tübingen, Germany) on September 9, 2024, had a pH value of 7.94. The dissolved organic carbon (DOC) measured as non-purgeable organic carbon (NPOC), was determined to be 11.6 mg/L. Dissolved iron and manganese concentrations were both below the detection limit of 0.04 mg/L (Mn) and 0.02 mg/L (Fe). Supplementary Table S6 summarizes the results of the wastewater sample characterization using IC with conductivity detection and MP-AES (for analytical methods see the Supplementary Information). Supplementary Table S7 contains additional information on a 24 h-mixed wastewater sample monitored by the WWTP Lustnau. The sampling point for the latter lies after the screen but before the grit chamber. Due to heavy rainfall the day and night before wastewater sampling the wastewater volume treated in the WWTP in Lustnau (Tübingen, Germany) on September 9, 2024, amounted to approximately 80 000 $m^3/d$ compared to 25,000–30,000 $m^3/d$ on an average dry day.

## Data availability

The data supporting the findings of this study is provided in the Supplementary Information. Additional data related to the study is available upon request from the corresponding authors.

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

## Acknowledgements

We thank Nora Simon, Hannah Arnold and Sara Wild (all University of Tübingen) for assistance in the laboratory, especially for sample derivatization. Furthermore, we thank Volker Karius from the University of Göttingen for conducting the X-ray diffractograms of the used

manganese dioxide and Markus Kramer from the University of Tübingen for conducting the NMR measurements. Mathis Athmer thanks The Chemical Industry Fund (FCI, Fonds der Chemischen Industrie, Germany) for his Kekulé fellowship. This research was financially supported by the Germany Research Foundation (BU 782/2-1, D.B.; HA 3453/17-1, S.B.H.) and the University of Tübingen (PRO-MARTIN-2023-11; P.R.M).

## Author contributions

A.M.R.: Conceptualization, Methodology, Validation, Formal Analysis, Investigation, Writing – Original Draft, Visualization, Project administration. P.R.M.: Conceptualization, Methodology, Validation, Funding acquisition, Writing – Review & Editing. M.A.: Methodology, Investigation, Formal Analysis, Writing – Review & Editing. S.B.: Investigation, Formal Analysis, Writing – Review & Editing. D.B.: Conceptualization, Funding Acquisition, Writing – Review & Editing. U.K.: Supervision, Writing – Review & Editing. C.H.: Conceptualization, Writing – Review & Editing. T.C.S.: Supervision, Writing – Review & Editing. S.B.H.: Conceptualization, Supervision, Resources, Writing – Review & Editing

## Funding

## Competing interests

The authors declare no competing interests.
