## [Transparent Peer Review file · Nature Communications]

Glyphosate is a transformation product of a widely used aminopolyphosphonate complexing agent

Corresponding Author: Dr Philipp Martin

Version 0:

Reviewer comments:

Reviewer #1

(Remarks to the Author)

The manuscript investigated the abiotic degradation of DTPMP by Mn-dioxide minerals. The formation of glyphosate highlighted another possible source of glyphosate other than agricultural application of glyphosate-based herbicides. This is the first study demonstrating the formation of glyphosate from DTPMP via abiotic oxidation by naturally existing minerals (MnO₂). The study is straightforward and easy to follow. However, some major and minor issues need to be resolved or justified before publication.

Major suggestions:

1. Is the studied concentration level of MnO₂ within the range of Mn minerals in the natural environments or in the wastewater treatment systems? If so, as determined in this study, the mol % of glyphosate that is produced from abiotic degradation of DTPMP by MnO₂ is very low (up to 0.16 mol %). Is this comparable to the biotic degradation of DTPMP?
2. The manuscript is lack of sufficient discussions on the mechanisms that lead to the observed results. For example, what is causing a lower formation rate of glyphosate but not AMPA under anoxic conditions? Potential impact of DTPMP:Mn ratios on the composition of degradation products or maybe even on the degradation pathways?
3. The study is also lack of sufficient investigation into the degradation process.
 - a. Since DTPMP and many of its intermediate products are oxidatively degraded, MnO₂ should be reduced, but there is no analysis to confirm this Mn reduction.
 - b. In addition, O₂ might have oxidized the reduced Mn(II) to Mn(III), which is also catalyzed by MnO₂, as proposed by the authors; however, there is no analysis to confirm this formation of Mn(III) phases in the solid phase.
 - c. Moreover, a much higher degradation rate was observed at 1 g/L MnO₂ than 0.1 g/L, is it still higher after normalizing the rate to the specific surface area of the mineral?
 - d. A rough calculation: 0.1 g/L MnO₂ is about 1.1 mM MnO₂, if the mineral is pure without any other elements. Is this amount sufficient to accept enough electrons to completely oxidize 1 mM DTPMP and some of its intermediate products? This may also provide us some insights into the potential role of O₂ during the degradation process.

Detail comments:

1. Figure 1: Based on the structure of DTPMP, it seems that cleavage of the red C–P bond or the C–P bonds on the other side of the molecule, instead of the labeled i) C–P bond, can also generate the same product. How to differentiate which C–P bond was cleaved? In addition, a proposed degradation scheme showing all measured and possible degradation products will be helpful for the audience to understand the degradation pathways.
2. Line 54: another recent publication showing glyphosate as a degradation product of APPs: Drzyzga, D; and Lipok, J. (2017) Analytical insight into degradation processes of aminopolyphosphonates as potential factors that induce cyanobacterial blooms. *Environ. Sci. Pollut. Res.* 24: 24364-24375.
3. Line 109: pH: what is the general pH range of wastewater in the treatment plants? How about neutral and slightly alkaline pH conditions?
4. Line 147: suggest a detailed discussion on the impact of DTPMP:Mn ratio on the composition/abundance of different major degradation products. It seems that a higher ratio may favor the accumulation of some intermediate products.
5. Line 151: the concentration of unidentified TP X seems to be similar or higher than orthophosphate in Figure 3. Any other analysis, such as 2-D NMR or tandem MS, to confirm the identity of this compound?
6. Figure 4: Concentration of AMPA is way much higher than glyphosate. Is it due to rapid degradation of glyphosate to AMPA? Or is it possible that AMPA directly formed, bypass the formation of glyphosate?

7. Line 201 and Table 2: why less glyphosate formed under anoxic conditions? Less degradation from its precursor compounds? However, if glyphosate is the necessary precursor of AMPA, less formation of AMPA in anoxic conditions should also be observed. Does this suggest that glyphosate is not a necessary precursor to AMPA?
8. Line 214: if Mn(III) formed on mineral surface, other analytical techniques can be used to confirm the change in oxidation state of Mn in the solid phase or the formation of a new Mn(III)-mineral on the surface of MnO₂?
9. Line 324: use of the buffer: does the buffer interact with MnO₂ as well? Is the pH change after the reaction monitored?

Reviewer #2

(Remarks to the Author)

The authors of this manuscript were members of a team that published a paper in *Water Research* (ref 14) in October 2024 suggesting that municipal wastewater treatment plants in Europe are important, but under-recognized sources of glyphosate. They also proposed that aminopolyphosphonates, mainly from laundry detergents are transformed to produce glyphosate. In this manuscript, the authors propose that manganese in wastewater is responsible for the conversion of the aminopolyphosphonates diethylenetriamine penta(methylene phosphonate) (DTPMP) into glyphosate through a reaction pathway that has a relatively low yield (i.e., <0.2%). Overall, I found the hypothesis of the authors to be interesting and the topic of the research to have substantial implications for efforts to control glyphosate pollution. Recognizing that this is an initial communication, it seems reasonable that some of the details needed to fully characterize these reactions might not be available. Nonetheless, I remain unconvinced that the authors have proven that this reaction is the source of the glyphosate. Given the potential significance of the findings of this study, I believe that a more thorough analysis of the processes and a better justification of the experimental conditions is needed prior to publication.

My specific concerns about the manuscript are detailed below.

1. If I understand the hypothesis correctly, the authors set out to determine if the reaction between DTPMP and Mn takes place between the location where it is first used (e.g., in laundry washing machines) and the discharge of a conventional wastewater treatment plant. The experiments described in the manuscript do not give me confidence that the reaction has been studied under environmentally relevant conditions as the authors claim. My first concern is that most chelating reagents like DTPMP will exist as complexes with metals. Have the authors proven that glyphosate is produced when the DTPMP is complexed by Fe(III), Ca(II) or other metals that are present in sewage?
2. Similarly, the authors chose to study extremely high concentrations of a crystalline manganese oxide. The form of the manganese oxide is known to play a very significant role in the transformation of organic compounds. I was disappointed in the supporting information, which did not give data on the polymorph of Mn oxide. Moreover, I do not understand the justification for assuming the presence of 100 or 1000 mg/L of pure Mn oxides in wastewater. The suspended solids in raw sewage or activated sludge mainly consist of organic matter with relatively low concentrations of minerals, like silica and iron oxides. Even under the unlikely situation in which sewage were to contain high concentrations of manganese oxides, it is likely that the surfaces would be covered with protein, biopolymers, surfactants and other substances that could block or slow the interactions of DTPMP with the surface. Under some conditions, sulfide might also form other Mn-containing minerals (e.g., MnS). Can the authors better justify the concentrations of Mn oxides used in these experiments? Do these reactions take place in the presence of other substances present in sewage?
3. The experiments were conducted at pH 6 using MES as a buffer. Oxidation reactions involving Mn-oxides tend to exhibit a strong pH dependence. Were these reactions replicated at higher pH values (e.g., pH 7-8 are typical of sewage and treated wastewater)?
4. The reaction between MnO₂ and DTPMP appears to have been studied by other authors. The formation of glyphosate as a minor transformation product seems to be thermodynamically possible and the fact that it is a minor pathway could help explain the fact that this reaction has not been considered by others. However, the lack of a quantitative treatment of the observed formation of glyphosate leaves the reader wondering whether the explanation is valid. If data are available on DTPMP concentrations in sewage and wastewater treatment plants, it should be possible to convert the observed losses of the parent compound into glyphosate concentrations formed in the process by using the yield measurements. Have the authors made these kinds of calculations? If so, can they explain the observations reported in the *Water Research* paper?
5. Consideration of the fate of DTPMP in wastewater without discussion of the possible importance of biotransformation of the compound seems incomplete. Is this compound transformed in sewage or in sewage treatment plants? If so, could biotransformation explain the observed formation of glyphosate?
6. The discussion of reaction kinetics needs considerable improvement. For example, I found the description of pseudo-zero order kinetics to be quite confusing. Yes, one might observe zero-order kinetics for some catalytic reactions, but drawing lines based on one or 4-5 data points (i.e., those where the parent compound is present at concentrations above the detection limit and then calculating r^2 values and rate constants to four significant figures and attempting to interpret mechanisms is an over-reach.
7. The authors state that Mn²⁺ also oxidizes DTPMP but they did only included results from a single experiment conducted with 1 mM of DTPMP and 1 mM Mn²⁺. In this experiment products were only observed after about 5 days. I struggle to relate this very unrealistic experiment and the timeframes needed to see the formation of glyphosate to the overall hypothesis of the authors. Stating that Mn²⁺ and oxygen can produce glyphosate under environmentally relevant conditions, as stated in

the abstract, seems like another over-reach.

Reviewer #3

(Remarks to the Author)

The study shows for the first time that glyphosate can be produced in trace amounts by the reaction of an aminophosphonate with manganese oxides under neutral pH conditions. This is a remarkable result in the field of water pollution, as the origin of glyphosate in the environment has recently been the subject of debate in Europe. The context is well described and supported by appropriate, up-to-date references. This study clearly shows that glyphosate can be produced by the transformation of a widely used non-toxic organic compound, and not just by its use in agriculture.

Glyphosate is not the main transformation product, which explains why it has been overlooked until now. Thus, specific analytical methods have been used in this study for its analysis at trace level. Results also show that concentrations of manganese oxide and dissolved oxygen strongly influence the yield of glyphosate formation and that glyphosate can also be formed in presence of dissolved manganese ions and oxygen for long reaction time. Interesting result also is the stability of glyphosate in presence of an excess of manganese oxide.

A reaction scheme is proposed based on literature data, which will need to be revised slightly (see remark below). The data are of high quality and obtained with the appropriate protocols and analytical methods. Protocols and methods are generally well described.

The authors suggest that this reaction can occur in wastewater treatment plants because aminophosphonates are used as complexing agents with detergents and manganese oxides are present in sewage sludge. However, the authors do not demonstrate that this reaction actually occurs in wastewater treatment plants. The experiments were carried out with a high concentration of aminophosphonate (1 mM) in pure aqueous suspensions of manganese oxides. The lower concentration of aminophosphonate and the presence of biomass and dissolved organic matter in a real wastewater treatment plant probably greatly reduce the significance of this reaction. This is a point that should be discussed further or possibly illustrated by carrying out new experiments, for example in the presence of biomass and/or dissolved effluent organic matter. It is well known that organic matter can adsorb onto manganese oxides and strongly inhibit their reactivity.

Below are some detailed comments.

1. The mechanism in Figure 1 needs to be revised slightly. If the initial cleavage takes place on the C-P bond, a methyl group must remain on the nitrogen, so the amine remains a tertiary amine and not a secondary amine as shown. In fact, the initial step would involve cleavage of the C-N bonds at both nitrogen atoms. These concomitant cleavages probably explain the small quantities of glyphosate formed during this transformation, compared with IDMP that is directly produced from one C-N bond cleavage.
2. Proposed protocol also involves the formation of an aldehyde compound as intermediate. This compound can be detected using the Fmoc derivatization method. Have any tests been carried out to identify this compound, for example by LC-HRMS?
3. Line 81. Meaning of IDMP is missing
4. Line 302. Unit for conductivity is $\mu\text{S}/\text{cm}$ not $\mu\text{S}/\text{cm}^2$.
5. Line 315. I assume it should read "was purchased" instead of "and purchased".
6. Line 319. Indicate in the protocol how MnO_2 suspensions were mixed during experiments or give reference.
7. Line 320. Falcon® instead of falcon
8. Line 327. It is stated here that due to the high sorption of DTPMP, the residues were desorbed by adding sodium phosphate and sodium hydroxide. It is not clear what the residues are here, i.e. residual DTPMP or all transformation products, including AMPA and glyphosate. In other words, please state whether the desorption procedure was used prior to any analysis of the transformation products. If so, how does this procedure affect the LOD and LOQ and what is the proportion of sorbed TPs?
9. Table 2. Typo: LOQ for glyphosate is $0.198 \mu\text{M}$ not $0.189 \mu\text{M}$.

Version 1:

Reviewer comments:

Reviewer #1

(Remarks to the Author)

The authors have done a great job answering my concerns/questions and revising the manuscript. Though my concerns on reduction and transformation of Mn-minerals are not resolved in this revision, I agree that these analyses are not the main focus of this study and can be done in future studies. Overall, the manuscript is well improved.

Reviewer #2

(Remarks to the Author)

I appreciate the efforts that the authors have made to address comments made by the three reviewers and am happy to see that they have included some additional experiments to address several of my main concerns about the extrapolation of the data in laboratory experiments to conditions in the environment.

I recognize that the intention of this manuscript is to demonstrate a proof of concept that MnO_2 can oxidize DTPMP under conditions encountered in sewage but I am still concerned about the issue of the MnO_2 polymorphs and their concentration.

If the editor is comfortable with the publication of a result that is still somewhat speculative I would not object, as long as the authors are more direct about the nature of the extrapolation that they have made. With respect to the concentration of Mn-oxides present in the samples, I accept the idea that wastewater solids (e.g., sludge or mixed liquor suspended solids) may contain 1 g/kg of Mn and that these experiments may approximate the concentrations detected in wet wastewater sludge (as the authors indicate in their response to my comment but this is not the question that the authors have set out to answer nor is it the meaningful question for the research. Sewage consists of a small amount of wastewater solids suspended in a relatively large volume of water. For example, if the sewage contained 1 gram of solids per liter of water then these experiments would have been conducted with Mn concentrations that are three orders of magnitude higher than those that are encountered in the environment. Perhaps biogenic Mn-oxides are much more reactive than the particles used in these experiments, but a better acknowledgment of this limitation is needed.

Similarly, I still find it surprising that the authors have been so casual about the form of Mn-oxides used in the experiments. It is hardly surprising that some random Mn oxide on the shelf of the lab included a few different crystalline phases and some amorphous oxides when analyzed by XRD. The presence of amorphous phases does not convince me that this is a surrogate for biogenic Mn-oxides. The absence of any data on where the Mn oxide originated (if it is from a mineral collection) or how it was synthesized (if it is synthetic) is a minimum expectation for a scientific study. As it stands, the experiments reported here are irreproducible because the type of Mn-oxide remains undefined. If the editor is willing to publish this work without such information a clearer statement is needed indicating that only one type of Mn oxide was studied, that its provenance is uncertain and that Mn oxides in sewage might behave in a much different manner because they will form by different mechanisms and will be more intimately associated with organic matter is necessary.

The fact that glyphosate does not form when DTPMP is added to raw wastewater (without amending with Mn-oxides) is evidence that there is an issue that requires further study.

Reviewer #3

(Remarks to the Author)

This manuscript presents the results of a study showing that the transformation of EDTMP phosphonate by MnO₂ can be a source of glyphosate, a pesticide widely used in agriculture for weed control. The authors have made significant changes to the original version. The objectives of the study are clearer and the new experiments with wastewater reinforce its relevance (but certain characteristics of wastewater are not usual, see below). The authors have responded very carefully to all the reviewers' comments and made the appropriate corrections.

However, there are still minor issues that need to be addressed before possible publication.

Minor detailed comments:

Line 232. Yield of glyphosate is 0.03 mol% (corresponding to 0.3 μM) not 0.03 μM.

Line 238. Please rephrase "Glyphosate and AMPA concentrations only rose after 1.5 respectively 1 hour, even though DTPMP" by, for example, "Glyphosate and AMPA concentrations only rose after 1.5 and 1 hour, respectively, even though DTPMP..."

Line 240. I would change "The maximum AMPA and glyphosate concentrations formed differed between the four experiments » by "The maximum concentrations of AMPA and glyphosate formed differed between the four experiments"

Line 285. In "Glyphosate and AMPA yields were up to 0.06 mol-% (glyphosate) and 10.3 mol-% (AMPA) after 168 hours", I would specify that it is in waste water.

Table 8. The effluent conductivity, ~100 mS/cm (i.e. 66 g NaCl/L equivalent), is particularly high for wastewater and given the ion concentrations in Table 7; this conductivity is probably not correct. Similarly, DOC (11.6 mg/L) seems relatively low for a wastewater influent (usual values for DOC are more between 100 and 300 mg/L in influent) and comparatively to the COD of 195 mg/L (which is low for influent but high for a treated waste water). Is this sample characteristic of a wastewater influent? Is this sample a wastewater influent or a treated wastewater effluent? Please clarify.

Supporting information

Caption of Figure S-5. The following sentence is not clear. Please rephrase. "Aqueous concentration of DTPMP quantified by means of IC-ICP-MS and total concentrations of glyphosate and AMPA quantified by means of LC-QQQ after derivatization in the replica the experiments over 96 hours using 1.0 g/L MnO₂ under oxic conditions."

AUTHOR RESPONSE TO REVIEWER COMMENTS

We would like to thank the reviewers for their very valuable comments which were insightful and helpful in improving our work and highlighting the novelty and relevance of our experimental results.

We performed additional experiments in the wastewater matrix that confirmed our earlier results obtained in pure water and substantially revised the text to address the two main points of criticism of the reviewers:

- i) The relevance of glyphosate formation from DTPMP under technically/environmentally relevant conditions*
- ii) Unclear formulation of the objectives and scope of the study*

Our new data on glyphosate formation in the wastewater matrix strengthen our earlier conclusions that manganese-driven oxidation of DTPMP is a potentially relevant source of glyphosate in technical and environmental systems. This is further supported by additional data on oxidation by dissolved $Mn^{2+}_{(aq)}/O_2$, which showed that glyphosate is formed from DTPMP even in the absence of the strong oxidant MnO_2 .

In regard to the scope of our study, we have emphasized that it is a proof-of-concept work. We reworded those parts of the manuscript where it could be assumed that we were attempting to elucidate the underlying reaction mechanisms in detail. Accordingly, we have removed discussions of potential or unidentified transient products of DTPMP transformation. Finally, we clarified that although the study provides experimental evidence that glyphosate forms from manganese-driven DTPMP transformation under environmentally relevant conditions, it cannot be used to quantitatively assess de novo glyphosate formation and inputs to European surface waters.

In the following section, we respond to the comments and, where appropriate, present the corresponding amendments in the revised version of the manuscript. The comments of the reviewers are presented in black font, while our responses are in italics and the revisions are in red font and indented.

Reviewer #1 (Remarks to the Author):

The manuscript investigated the abiotic degradation of DTPMP by Mn-dioxide minerals. The formation of glyphosate highlighted another possible source of glyphosate other than agricultural application of glyphosate-based herbicides. This is the first study demonstrating the formation of glyphosate from DTPMP via abiotic oxidation by naturally existing minerals (MnO_2). The study is straightforward and easy to follow. However, some major and minor issues need to be resolved or justified before publication.

Major suggestions:

1. Is the studied concentration level of MnO_2 within the range of Mn minerals in the natural environments or in the wastewater treatment systems? If so, as determined in this study, the mol % of glyphosate that is produced from abiotic degradation of DTPMP by MnO_2 is very low (up to 0.16 mol %). Is this comparable to the biotic degradation of DTPMP?

Response:

We would like to thank the reviewer for her/his insightful comments. In the following we will address separately:

a) Manganese concentrations

Our work aims to provide experimental evidence for the formation of glyphosate from DTPMP in the presence of manganese. As such, it is designed as a proof-of-concept study that sets the stage for follow-up studies addressing more specific objectives, such as the extent to which such processes can explain elevated glyphosate concentrations observed in surface waters.

The mere comparison of total Mn or MnO_x contents in natural or technical environments with the MnO_2 mineral concentrations used in this study to judge on its environmental relevance is challenging. This is due to the importance of MnO_x surface areas for the mineral's reactivity. Mn oxides often occur as fine-grained particles or coatings in the environment¹, which increases the available surface and therefore the reactivity. In our case, we used pure Mn oxide. Nevertheless, we compiled total Mn contents in natural and technical environments to contextualize our concentrations:

The range of Mn concentration in wastewater sludge is 600 – 1,500 mg/kg respectively 3,300 – 3,900 dry mass according to the German environment agency² resp. Milik et al. (2017)³. The moisture content in sewage sludge is typically ≥ 90 %, depending on the type of sludge³.

Upon the assumption of 95 % moisture content, Mn concentrations regarding the wet mass are in the range of 30 to 195 mg/kg. Hence our lower MnO₂ concentration (0.1 g/L) matches the range of total Mn in wet wastewater sludge. The total Mn content in topsoils typically ranges between 20 and 3000 mg/kg⁴ with typical levels around 600 mg/kg⁵. The porosity of hyporheic zone sediments is usually in the range of 0.3 to 0.4, thus by assuming a porosity of 0.35 we arrive at about 400 mg/kg wet mass. The equilibrium between MnO_x minerals and Mn²⁺ in solution depends on the pH, redox conditions (Mn^{III} and Mn^{IV} oxides are generally more stable at higher pH values and under oxidizing conditions⁶ and in the presence of Mn²⁺ oxidizing micro-organisms⁷).

b) Biotransformation

Knowledge on the biotransformation of aminopolyphosphonates (APPs) like DTPMP is still scarce. Few studies investigated APP biotransformation, but the presence of manganese in all bacterial growth media and the observation of APP transformation in abiotic controls (DTPMP in the pure medium without bacteria) or the lack of sufficient negative controls impairs the conclusion that the microbes are using the DTPMP as (sole) phosphorus source. Rather, it seems conceivable that the microbes use PO₄³⁻-P or AMPA-P resulting from abiotic Mn-driven transformation of DTPMP to facilitate growth, due to the presence of Mn²⁺ and O₂. The abiotic transformation of aminotrismethylenephosphonate (ATMP) by Mn²⁺/O₂ was already investigated in detail in previous studies⁸⁻¹⁰. It could be shown that even if the Mn²⁺:ATMP ratio is as low as 1:100, ATMP is completely transformed after approximately 30 hours. Negative controls from i) Steber & Wierich (1987)¹¹, ii) Schowanek & Verstraete (1990)¹² and iii) Drzyzga & Lipok (2017)¹¹ showed chemical ATMP (i and iii) and GBMP (iii) and DTPMP (ii and iii) transformation in non-inoculated growth media in the dark, while Riedel et al. (2023 and 2024a and 2024b)¹³⁻¹⁵ miss cell free controls of DTPMP in the used bacterial growth medium. This raises significant questions regarding the occurrence of the actual extent of DTPMP biotransformation in these studies.

The formation of glyphosate from DTPMP biotransformation has never been reported anywhere until now.

However, thanks to the reviewers' comments, we realized there was a gap in our manuscript. In the revised version of the manuscript, we amended the conclusion with a small paragraph on biotic APP degradation and the role of manganese in the bacterial growth medium. It now reads:

Lines 346 to 354:

In addition, our findings may also provide new clues for revisiting APP biotransformation studies. All bacterial growth media used in published APP

biotransformation studies contain dissolved manganese^{53,57-61}. Thus, manganese-driven oxidation may occur in parallel with or instead of biotransformation of DTPMP and other APPs. Martin et al (2022)¹⁰ showed that even at a molar ratio of 1:100 (Mn:ATMP), ATMP was completely degraded within 30 hours. Thus, it is conceivable that in biotransformation studies under oxic conditions in the presence of dissolved Mn²⁺, microorganisms may utilize chemical transformation products of APPs, such as phosphate, IDMP, or AMPA, as a phosphorus source rather than or in addition to directly metabolizing the target APPs.

2. The manuscript is lack of sufficient discussions on the mechanisms that lead to the observed results. For example, what is causing a lower formation rate of glyphosate but not AMPA under anoxic conditions? Potential impact of DTPMP:Mn ratios on the composition of degradation products or maybe even on the degradation pathways?

Response:

We agree with the reviewer that we do not discuss the mechanism in detail. As mentioned earlier, this manuscript is an initial communication on the observation that glyphosate can form from DTPMP via an environmentally relevant oxidation process. A part of our team investigated the mechanism of IDMP (as a model compound for APPs) on MnO₂ in detail (Röhnelt et al. 2023¹⁶, see response to comment 3 b). However, the focus of our current study is on the formation of the critical and so far overlooked product glyphosate. The identification of other transformation products/intermediates, which are important for deciphering a detailed reaction mechanism and pathway, is beyond the scope of this study. The elucidation of the reaction mechanisms of the oxidation of DTPMP in heterogeneous and homogeneous solution and the identification of the numerous transient products is extremely challenging and a study in itself.

Thanks to the reviewers' comments, we realized during the revision process that we put too much emphasis on further transformation products like X, Y, Z, without going into more detail or identifying their chemical structures, and as a result leaving the reader with questions. As the identification of those compounds and – especially – the elucidation of the transformation mechanism would require quite different analytical methods (e.g., high-resolution mass spectrometry (HRMS), compound-specific stable isotope analysis (CSIA), Fourier-transform infrared spectroscopy (FTIR)) and further experiments (e.g., isotope-labelling, different concentration set-ups, different minerals), this would be beyond the scope for this “proof-of concept” study.

Consequently, we focus more on glyphosate and AMPA formation in the revised version of the manuscript. The additional experiments conducted and added in a new sub-section, also focus on DTPMP, glyphosate and AMPA solely. Furthermore, we emphasized that the identification of intermediates and elucidation of the reaction mechanism is a great and challenging task to tackle in future work. The new text parts now read as follows:

Lines 180 to 200:

During DTPMP transformation, the formation of various phosphorus-containing TPs was monitored in the aqueous phase using ion chromatography (IC) coupled to inductive-coupled plasma mass spectrometry (ICP-MS). Fig. 3 shows an exemplary chromatogram (1 g/L MnO₂, anoxic conditions, reaction time of 2 h, aqueous phase) next to a mix-standard of 30 ppb P (0.97 μM) per compound. The main TPs identified based on retention times and reference compounds in all experiments were IDMP and phosphate, consistent with previous studies^{16-18,25}. Based on the initial DTPMP concentration, IDMP formation reached up to 97 mol-% (0.1 g/L MnO₂, anoxic conditions), while PO₄³⁻ formation peaked at 153 mol-% (1.0 g/L MnO₂, oxic conditions). Regarding phosphate, the maximum molar yield amounts to 500 mol-%, due to DTPMP's five phosphonate moieties. DTPMP, IDMP, and PO₄³⁻ concentrations over time in the aqueous phase are depicted in supplementary Fig. S4.

In the exemplary chromatogram shown in Fig. 3, a double peak is visible at the retention time of AMPA (83.0 s), while a triple peak is observed around the RT of glyphosate (167.5 s). Both peaks, even if considered to be the respective compounds, were below the LOQs (0.9 ppb P for AMPA and 1.7 ppb P for glyphosate) and therefore represent an almost negligible fraction within the TP spectrum. Thus, it is not surprising that the formation of glyphosate has so far been mostly overlooked.

Lines 339 to 345:

Overall, this study for the first time provides experimental evidence for conversion of a widely used non-toxic commodity compound into a highly debated pesticide¹⁷⁻²⁰ under environmentally relevant reaction conditions. While we could demonstrate that the reaction is chemically feasible in the laboratory, future research should elucidate in detail how environmental conditions affect glyphosate formation from DTPMP and related APPs, the formation and identification of key intermediates and field studies including yields in wastewater treatment plants.

3. The study is also lack of sufficient investigation into the degradation process.

a. Since DTPMP and many of its intermediate products are oxidatively degraded, MnO_2 should be reduced, but there is no analysis to confirm this Mn reduction.

Response:

We agree with the reviewer on the mechanistic issue. However, in this study we did not set out to analyze/quantify Mn^{2+} in solution because i) we know it is formed and ii) interpretation of the $\text{Mn}^{2+}_{\text{aq}}$ data is not straightforward. We will explain this in detail below:

In a former study, a part of our team investigated the transformation of IDMP on manganese dioxide, in which we elucidated Mn^{2+} formation¹⁶. We showed that in accordance with the literature¹⁷, Mn^{2+} was formed. In the presented study, direct DTPMP oxidation by MnO_2 was also visible to the eye as MnO_2 dissolution in the reaction suspensions, as the mineral concentration was visibly depleted and the suspension changed its color from colorless to yellowish/brownish toward the end of the experiments. Thus, it is justified to presume that MnO_2 is reduced and Mn^{2+} is formed in our DTPMP experiments.

We now emphasized in the revised manuscript that generally Mn^{2+} forms upon Mn^{IV} reduction by organic compounds, hence the revised paragraph now reads:

Lines 150 to 157:

The rate enhancing role of oxygen may be related to multiple processes involving different redox states of manganese as reported in literature. Under oxic conditions, Mn^{2+} (formed by the reduction of Mn^{IV} ^{26,30}) is known to catalyze DTPMP (and ATMP, EDTMP) oxidation by O_2 ($\text{Mn}^{2+}/\text{O}_2$) in homogeneous solution^{28,34}. However, the much slower reaction kinetics of DTPMP in homogeneous solution (1 mM Mn^{2+} and O_2) compared to the heterogeneous systems found in this study clearly show that the strongly enhancing role of O_2 in MnO_2 experiments must be due to processes other than mere $\text{Mn}^{2+}/\text{O}_2$ interaction, probably involving the formation of Mn^{III} on the mineral surface.

Although we appreciate the comment, we feel confident that the quantification of formed aqueous Mn^{2+} does not help answer our research questions, which rather refer to the formation and yield of glyphosate and AMPA and less on the mechanism. The elucidation of the reaction mechanism involving the organic compounds, investigating the redox reaction from the inorganic site (Mn^{II} , Mn^{III} , Mn^{IV}) would require different experiments and analytical techniques, such as, e.g., FTIR and X-ray photoelectron spectroscopy (XPS).

Additionally, we want to point out the difficulties of quantifying and interpreting the amount of formed Mn^{2+} in the investigated system. In a previous work (Röhnelt et al. 2023¹⁶) a part of

our team calculated the total amount of Mn^{2+} released from IDMP oxidation by MnO_2 – aqueous and sorbed, as Mn^{2+} can sorb to the mineral surface – via i) the aqueous Mn measured in each experiment and ii) a sorption isotherm recorded in a buffered solution without IDMP. This was – of course – an approximation, as recording a sorption isotherm of Mn^{2+} on MnO_2 in the IDMP matrix is not possible, as IDMP is oxidized, MnO_2 reduced and more Mn^{2+} released continuously. However, with that approximation we found roughly one Mn^{2+} released for each IDMP molecule transformed.

DTPMP as a precursor increases the complexity, as several reactions occur simultaneously.

1. A plethora of transformation products forms, which are partly further transformed themselves.
2. DTPMP (and plausibly also some transformation products) also reacts with Mn^{2+}/O_2 in solution.
3. MnO_2 acts catalytically in addition to its direct oxidation capacity (see responses below).
4. Mn^{2+} itself will undergo sorption onto the MnO_2 to a certain extent. Yet, estimating the amount of sorbed Mn^{2+} here based on the sorption isotherm in a buffered solution without the polyphosphonate matrix is too much of an outreach in our eyes.

Thus, the interpretation of Mn^{2+}_{aq} data would be quite inconclusive. This is the rationale behind the decision not to invest effort in determining the concentration of Mn^{2+}_{aq} while designing the experiments.

b. In addition, O_2 might have oxidized the reduced Mn^{II} to Mn^{III} , which is also catalyzed by MnO_2 , as proposed by the authors; however, there is no analysis to confirm this formation of Mn^{III} phases in the solid phase.

Response:

This is correct, we did not analyze any mineral phases after the reaction. The reason for this is that we do not focus on the heterogeneous oxidation mechanism in this study and do not intend to explain it in detail. However, thanks to the reviewers' comment we realized that we should elaborate in more detail on possible underlying processes. As we observed a clear dependency of the transformation kinetics on the presence of oxygen, we name/discuss in the revised manuscript three different plausible processes well described in literature, how oxygen interacts with MnO_2 and could potentially influence (increase) DTPMP oxidation by MnO_2 . The text now reads:

Lines 150 to 162:

The rate enhancing role of oxygen may be related to multiple processes involving different redox states of manganese as reported in literature. Under oxic conditions, Mn^{2+} (formed by the reduction of Mn^{IV} ^{26,30}) is known to catalyze DTPMP (and ATMP, EDTMP) oxidation by O_2 (Mn^{2+}/O_2) in homogeneous solution^{28,34}. However, the much slower reaction kinetics of DTPMP in homogeneous solution (1 mM Mn^{2+} and O_2) compared to the heterogeneous systems found in this study clearly show that the strongly enhancing role of O_2 in MnO_2 experiments must be due to processes other than mere Mn^{2+}/O_2 interaction, probably involving the formation of Mn^{III} on the mineral surface.

Manganese minerals with elevated Mn^{III} content appear to be more reactive oxidants^{30,35,36}. In heterogeneous systems containing Mn^{2+} and MnO_2 , Mn^{III} can be formed by comproportionation and be associated with the mineral surface or reside in solution³⁵. Furthermore, MnO_2 can catalyze the oxidation of Mn^{II} by O_2 to Mn^{III} ³⁷. Finally, MnO_2 itself may act as a direct oxidant but also as a catalyst in connection with O_2 ³⁸.

In a previous study (Röhnelt et al. 2023¹⁶), a part of our team intensively investigated the heterogenous oxidation mechanism of IDMP by MnO_2 . In the study presented here, on the contrary, we conducted a proof-of concept study on the formation of glyphosate from the manganese-driven oxidation of DTPMP, which main research questions are:

- I) *Can glyphosate form from DTPMP via an environmentally relevant oxidation process (MnO_2 or Mn^{2+}/O_2)?
 - a. *Does this formation take place under anoxic and oxic conditions?**
- II) *Which maximum molar yields of glyphosate and AMPA can be achieved via this process?*
- III) *Which effect has an environmentally relevant matrix (wastewater) on glyphosate & AMPA formation?*

c. Moreover, a much higher degradation rate was observed at 1 g/L MnO_2 than 0.1 g/L, is it still higher after normalizing the rate to the specific surface area of the mineral?

Response:

We thank the reviewer for that important suggestion. We normalized the pseudo-0th-order reaction rates to the available mineral surface (m^2/L) and added the normalized degradation rates to Table 1. The normalized reaction rates showed a slightly different trend than the non-

normalized ones. Still, the oxic experiments are fastest, but the lower concentrated mineral suspensions (0.1 g/L) show higher normalized reaction rates than the higher concentrated ones (1.0 g/L). Those insights indicate the involvement of catalytic activity of MnO₂ next to direct oxidation by MnO₂. Direct oxidation by MnO₂ was also optically visible in the reaction suspensions, as the mineral concentration was visibly depleted and the suspension changed its color from colorless to yellowish/brownish toward the end of the experiments. The text now reads:

Lines 146 to 151:

Normalization of the reaction rate constants to the specific surface area (k_{norm}) provided further insights into the role of oxygen and available surface area. k_{norm} -values showed that the oxic experiments were faster compared to their anoxic counterparts.

The rate enhancing role of oxygen may be related to multiple processes involving different redox states of manganese as reported in literature.

Table 1 Pseudo-0th order reaction rate constants (k) and those reaction rate constants normalized to the surface area (k_{norm}) for DTPMP transformation in the four experiments with MnO₂ in MES buffer. The standard errors of the linear regression are given as $\pm x$. R^2 is the regression coefficient of the linear regression from the start of the experiment until complete DTPMP transformation (24 h excluded for 0.1 g/L anoxic due to low data point density). The time interval denotes the time interval/timepoints included in the linear regression.

c(MnO₂)	1.0 g/L	0.1 g/L	1.0 g/L	0.1 g/L
O₂	oxic	oxic	anoxic	anoxic
k in $\mu\text{M}/\text{h}$	2729 \pm 800	650 \pm 106	946 \pm 26	133 \pm 3
k_{norm} in $\mu\text{mol}/(\text{m}^2\cdot\text{h})$	42 \pm 12	101 \pm 16	14.7 \pm 0.4	20.6 \pm 0.5
R²	0.842	0.859	0.996	0.988
Linear section in h	0-0.35	0-1.5	0-0.67	0-6.0

d. A rough calculation: 0.1 g/L MnO₂ is about 1.1 mM MnO₂, if the mineral is pure without any other elements. Is this amount sufficient to accept enough electrons to completely oxidize 1 mM DTPMP and some of its intermediate products? This may also provide us some insights into the potential role of O₂ during the degradation process.

Response:

We thank the reviewer for the comment and agree. The electron capacity of 0.1 g/L MnO₂ (1.1 mM), would certainly not be enough to oxidize DTPMP repeatedly to yield that number of intermediates and transformation products. In the revised manuscript including the normalized reaction rates to Table 1, it is shown that i) the normalized reaction rates of the oxic experiments are three to five times higher than those of the anoxic ones and ii) the normalized reaction rates of the 0.1 g/L MnO₂ experiments are higher than those of the 1.0 g/L experiments (see above). This indicates that MnO₂ also acts catalytically, as shown for the oxidation of toluene by MnO₂¹⁸. Based on the comment, we added the following discussion:

Lines 163 to 169:

Previous research on IDMP transformation using the same sample of MnO₂ demonstrated that roughly two electrons are accepted by MnO₂ per transformation of one IDMP molecule²⁶. Thus, the electron-accepting capacity of 0.1 g/L MnO₂, which corresponds to 1.1 mM MnO₂, cannot explain the complete transformation of 1 mM DTPMP to smaller transformation products, which are partially further oxidized. Moreover, as normalized reaction rate constants both under oxic and anoxic conditions were higher for low mineral concentrations (0.1 g/L vs. 1.0 g/L) a catalytic role of MnO₂ next to its direct oxidation activity is evident.

Detail comments:

1. Figure 1: Based on the structure of DTPMP, it seems that cleavage of the red C–P bond or the C–P bonds on the other side of the molecule, instead of the labeled i) C–P bond, can also generate the same product. How to differentiate which C–P bond was cleaved? In addition, a proposed degradation scheme showing all measured and possible degradation products will be helpful for the audience to understand the degradation pathways.

Response:

We agree with the reviewer that multiple C–N cleavages could lead to the same result, as the DTPMP molecule consists of recurrent moieties. Glyphosate can be formed from any phosphonate group in the DTPMP molecule. The purpose of this scheme is to show one

possible pathway of how to yield glyphosate. Based on the comment, we realized that we need to emphasize this in the caption of Figure 1. A discrimination of the position of the bond cleavage is not possible without isotope-labelling of specific atoms with the current analytical methodology available.

However, thanks to this comment and Reviewer #3 it became clear that we must clarify the cleavages shown in Figure 1. We have shown one C-N and one C-P cleavage in the scheme, because PO_4^{3-} is a major and widely known transformation product of DTPMP and all other phosphonates. However, for glyphosate to be formed we ultimately need two C-N bond cleavages, as glyphosate contains a secondary amine function. This is why we updated Figure 1 for the revised version of the manuscript, now showing two C-N bond cleavages plus terminal C oxidation to yield the formation of glyphosate.

We thank the reviewer for the suggestion to include more chemical structures. We decided to include AMPA, IDMP and PO_4^{3-} in Figure 1, as it is important to understand that AMPA will probably form primarily directly from DTPMP or via IDMP and is - in addition to that - a transformation product of glyphosate. Figure 1 and its caption now read:

Fig. 1 Schematic representation of the formation of phosphate, AMPA, IDMP and glyphosate (proposed) from DTPMP. Phosphate can form via one C-P bond cleavage (iv), IDMP is formed via one C-N bond cleavage (ii), while AMPA is formed via two C-N bond cleavages (i, ii). We propose one pathway for the formation of glyphosate from DTPMP via two C-N bond cleavages (i, iii) and oxidation of the terminal C first to the aldehyde (v) and then to the carboxylic acid (vi). The symmetry of the DTPMP molecule, which contains five phosphonate groups, allows multiple equivalent bond cleavages to lead to the same resultant product. For clarity, only one representative option for each potential cleavage is illustrated. All compounds are depicted in their fully deprotonated forms.

2. Line 54: another recent publication showing glyphosate as a degradation product of APPs: Drzyzga, D; and Lipok, J. (2017) Analytical insight into degradation processes of aminopolyphosphonates as potential factors that induce cyanobacterial blooms. Environ. Sci. Pollut. Res. 24: 24364-24375.

Response:

We thank the reviewer for the comment. However, upon thorough examination of the study by Drzyzga and Lipok (2017)¹¹, we can conclusively state that the research does not

demonstrate the formation of glyphosate as a metabolic product resulting from the degradation of DTPMP by cyanobacterial species (Table 6, p. 24372). The only APP from which 1.5 μM glyphosate were formed is GBMP. GBMP already possesses a carboxylic acid group, thus, to yield glyphosate, a C-N bond needs to be cleaved, but no oxidation to a carboxylic acid as in DTPMP transformation is necessary.

3. Line 109: pH: what is the general pH range of wastewater in the treatment plants? How about neutral and slightly alkaline pH conditions?

Response:

We thank the reviewer for that question. The general pH of untreated wastewater in WWTPs is mostly around 7.5 to 8¹⁹. We regard this comment and further questions below on the impact of metal cations, anions and NOM as very essential. In order to account for these comments, we conducted new experiments using sterile-filtered wastewater (pH 8) as our matrix. Both experiments started with 1 mM DTPMP, with either 1 g/L MnO_2 or 1 mM Mn^{2+} as oxidant/catalyst. The new subchapter “Formation of glyphosate and AMPA from DTPMP in wastewater matrix” now reads:

Lines 272 to 295:

To address the environmental relevance of the observations in pure water, experiments containing 1 g/L MnO_2 and 1 mM Mn^{2+} were conducted in wastewater (pH 8, sterile filtrated; see Materials & Methods for details on the wastewater sample). In control experiments with unspiked wastewater with and without MnO_2 or Mn^{2+} , negligible glyphosate and AMPA concentrations were occasionally detected (see Fig. S6). Dissolved manganese in the wastewater sample was below the detection limit (<0.04 mg/L).

DTPMP transformation kinetics with 1.0 g/L MnO_2 in the wastewater matrix were slower than in MES buffer at pH 6 (see Fig. S7 b) in line with a lower oxidation potential of manganese oxides at higher pH^{47,48}, as well as increased electrostatic repulsion between DTPMP and the mineral surface (point of zero charge of 5.6)^{30,49}. Furthermore, differences between the experiments with MES buffer and wastewater are likely due to the complex wastewater-matrix containing organics (such as other complexing agents) and cations (e.g., calcium), which were reported to influence APP transformation and sorption^{18,50}. Glyphosate and AMPA yields were up to 0.06 mol-% (glyphosate) and 10.3 mol-% (AMPA) after 168 hours. However, in wastewater spiked with 1 mM Mn^{2+} , DTPMP transformation kinetics were faster and had higher glyphosate and AMPA yields compared to MES-buffered pure water at pH 6 (see Fig.

5.) In wastewater spiked with 1 mM Mn²⁺, the highest glyphosate yield (0.42 mol-%) of all experiments conducted in this study was observed (see Table 2), albeit only after 240 h. Possibly, the reaction kinetics are faster at pH 8 due to stronger complex formation of Mn^{II} and DTPMP, such as shown for ATMP²⁸. At higher pH, less protons compete with metal ions present in solution and DTPMP is more negatively charged⁵¹. Furthermore, Mn^{III} complexes might play a role at higher pH. While the stability of Mn^{III} complexes varies depending on the ligand, certain ligands including desferrioxamine B show higher stability of Mn^{III} complexes at pH values between 7-11⁵².

4. Line 147: suggest a detailed discussion on the impact of DTPMP:Mn ratio on the composition/abundance of different major degradation products. It seems that a higher ratio may favor the accumulation of some intermediate products.

Response:

We thank the reviewer for the comment. This manuscript is a proof-of-concept study and focuses the formation of glyphosate and AMPA, not the elucidation of the mechanism. We realized that labelling and describing unknown transformation products raised many questions that we cannot answer yet. Therefore, we revised the subchapter “Formation of TPs from DTPMP” and omitted the lettering for unknown substances in Figure 3. It now reads:

Lines 180 to 200:

During DTPMP transformation, the formation of various phosphorus-containing TPs was monitored in the aqueous phase using ion chromatography (IC) coupled to inductive-coupled plasma mass spectrometry (ICP-MS). Fig. 3 shows an exemplary chromatogram (1 g/L MnO₂, anoxic conditions, reaction time of 2 h, aqueous phase) next to a mix-standard of 30 ppb P (0.97 μM) per compound. The main TPs identified based on retention times and reference compounds in all experiments were IDMP and phosphate, consistent with previous studies^{16-18,28}. Based on the initial DTPMP concentration, IDMP formation reached up to 97 mol-% (0.1 g/L MnO₂, anoxic conditions), while PO₄³⁻ formation peaked at 153 mol-% (1.0 g/L MnO₂, oxic conditions). Regarding phosphate, the maximum molar yield amounts to 500 mol-%, due to DTPMP's five phosphonate moieties. DTPMP, IDMP, and PO₄³⁻ concentrations over time in the aqueous phase are depicted in supplementary Fig. S4.

In the exemplary chromatogram shown in Fig. 3, a double peak is visible at the retention time of AMPA (83.0 s), while a triple peak is observed around the RT of glyphosate (167.5 s). Both peaks, even if considered to be the respective compounds, were below the instrumental LODs (0.9 ppb P for AMPA and 1.7 ppb P for glyphosate)

and therefore represent an almost negligible fraction within the TP spectrum. Thus, it is not surprising that the formation of glyphosate has so far been mostly overlooked.

To verify glyphosate and AMPA formation during DTPMP transformation, we used FMOc derivatization and subsequent quantification by means of reversed-phase high-performance liquid chromatography (RP-HPLC) coupled to a triple-quadrupole (QQQ) mass spectrometer, an established trace analysis method for glyphosate and AMPA³⁹⁻⁴¹.

Fig. 3 Phosphorus-selective IC-ICP-MS chromatogram of the aqueous fraction of duplicate A of the experiment containing 1.0 g/L MnO₂ under anoxic conditions, reaction time 2 h (see Fig. 2 c), overlaid by the chromatogram of a 30 ppb P standard mix including the denoted compounds (30 ppb P per compound). The sample was diluted 1:1000 to match the calibration range. Abbreviations for standard compounds not described in the text: 2-AEP = 2-aminoethylphosphonate, MPA = methylphosphonate, PAA = phosphonoacetic acid.

5. Line 151: the concentration of unidentified TP X seems to be similar or higher than orthophosphate in Figure 3. Any other analysis, such as 2-D NMR or tandem MS, to confirm the identity of this compound?

Response:

We thank the reviewer for the suggestion and agree that undoubtedly it is vital to further investigate the intermediates to better understand the formation pathway and mechanisms of glyphosate in the investigated system. However, as mentioned in the previous answer the (tentative) identification of intermediates by non-target/ suspected-target analysis requires a separate study and is beyond the scope of the presented work. Thus, we mentioned this important future work in the outlook of our manuscript:

Lines 341 to 345:

While we could demonstrate that the reaction is chemically feasible in the laboratory, future research should elucidate in detail how environmental conditions affect glyphosate formation from DTPMP and related APPs, the formation and identification of key intermediates and field studies including yields in wastewater treatment plants.

6. Figure 4: Concentration of AMPA is way much higher than glyphosate. Is it due to rapid degradation of glyphosate to AMPA? Or is it possible that AMPA directly formed, bypass the formation of glyphosate?

Response:

We thank the reviewer for the valuable input. We now clarified in the manuscript that AMPA formation can bypass glyphosate formation. AMPA itself is a well-known transformation product of APPs including DTPMP²⁰⁻²³. In addition, it should be noted that theoretically 3 mole of AMPA but only 2 mole of glyphosate can form from one mole DTPMP.

Lines 56 to 59:

Aminopolyphosphonates (APPs), which are widely used in laundry detergents in the EU¹⁶, are known precursors of aminomethylphosphonate (AMPA)¹⁷⁻²¹. Since the basic structure of glyphosate is already present in certain APPs (see Fig. 1), APPs are suspected precursors for glyphosate, too^{12,22}.

Lines 70 to 73:

Transformation of APPs (photolysis, Mn^{2+}/O_2 , $MnOOH$) with AMPA, iminodi(methylene phosphonate) (IDMP) and phosphate as major TPs (see Fig. 1) is well documented in the literature^{17-19,26-28}. However, evidence for glyphosate formation is limited to ozonation of EDTMP in drinking water²² [...].

To yield AMPA, merely two C-N bonds in DTPMP need to be cleaved (second bond cleavage on a different position than for glyphosate, see new Figure 1). The formation of glyphosate involves two C-N cleavages, too, followed by the dual oxidation of the terminal C – which is apparently quite unlikely to occur (see molar yields glyphosate). If glyphosate then is subject to another C-N cleavage, we would also arrive at AMPA. Thus, we assume that the monitored AMPA evolving via the “glyphosate pathway” is highly unlikely. This is further supported by the “long-term” stability of glyphosate concentrations, as glyphosate concentrations do

increase or stay constant over the course of the experiments (up to 240 h in a wastewater experiment). In the complex matrix of evolving transformation and intermediate products, glyphosate is present in comparably very low concentrations (see Figure 3) probably a rather small and weak complexing agent (one phosphonate group) compared to other transformation products (TPs) exhibiting two to four phosphonate groups. Therefore, it seems reasonable that the MnO₂ mineral surface (or Mn²⁺ in solution) is primarily complexed with other TPs and therefore not accessible for glyphosate.

Additionally, if glyphosate would be so instable, that it almost immediately transforms to AMPA and is just a very short-lived intermediate, we would not expect to see stable or even increasing glyphosate concentrations over days, while the AMPA concentrations are magnitudes higher from the very beginning.

Thanks to the reviewers' comments we realized that this needed to be emphasized in the text. The revised text now reads:

Lines 259 to 266:

This experiment demonstrates that AMPA and glyphosate – even in the most reactive suspension after 4 days – are not completely transformed. This is interesting, as the oxidation of glyphosate and AMPA on manganese oxides has been extensively studied and both compounds can be oxidized by MnO_x^{43–46}. Thus, further investigations are required to better understand the stability and further transformation of glyphosate and AMPA in consecutive reactions. The accumulation of TPs during the experiment creates a complex matrix that may impede the reaction between AMPA or glyphosate and MnO₂. These TPs potentially occupy the mineral surface, reducing the active sites available for further reactions.

Fig. 1 Schematic representation of the formation of phosphate, AMPA, IDMP and glyphosate (proposed) from DTPMP. Phosphate can form via one C-P bond cleavage (iv), IDMP is formed via one C-N bond cleavage (ii), while AMPA is formed via two C-N bond cleavages (i, ii). We propose one pathway for the formation of glyphosate from DTPMP via two C-N bond cleavages (i, iii) and oxidation of the terminal C first to the aldehyde (v) and then to the carboxylic acid (vi). The symmetry of the DTPMP molecule, which contains five phosphonate groups, allows multiple equivalent bond cleavages to lead to the same resultant product. For clarity, only one representative option for each potential cleavage is illustrated. All compounds are depicted in their fully deprotonated forms.

7. Line 201 and Table 2: why less glyphosate formed under anoxic conditions? Less degradation from its precursor compounds? However, if glyphosate is the necessary precursor of AMPA, less formation of AMPA in anoxic conditions should also be observed. Does this suggest that glyphosate is not a necessary precursor to AMPA?

Response:

We thank the reviewer for the comment. However, as explained in the previous comment, glyphosate is not a necessary precursor for AMPA – AMPA formation will most likely bypass

glyphosate formation. While AMPA formation just relies on two C-N bond cleavages, glyphosate formation additionally requires the dual oxidation of a terminal C to reach the carboxylic acid. Apparently, molecular oxygen is somehow involved in the formation of the carboxylic acid, while the C-N bond cleavages to reach AMPA do not require molecular oxygen. This might be connected to differing reaction pathways with/without the presence of molecular oxygen: If oxygen is present, the reaction rates for DTPMP transformation (see above and Table 1) are much higher. We conclude that this is due to the involvement of catalytic MnO_2/O_2 activity, next to the direct oxidizing capacity of MnO_2 .

We refer to our detailed answer on AMPA formation bypassing glyphosate.

We supplemented the introduction and discussion in the manuscript to make this clear to the reader.

Introduction:

New Figure 1 + caption, please see answer above.

Results and Discussion:

Lines 211 to 212:

While AMPA is the main TP of glyphosate in the environment^{7,11,42}, the main path for AMPA formation from DTPMP is via two C-N bond cleavages (see Fig. 1).

8. Line 214: if Mn^{III} formed on mineral surface, other analytical techniques can be used to confirm the change in oxidation state of Mn in the solid phase or the formation of a new Mn^{III} -mineral on the surface of MnO_2 ?

Response:

We thank the reviewer for this comment and copy our answer from above to a similar question from Reviewer 1: This is correct, we did not analyze any mineral phases after the reaction. The reason is that we do not aim at explaining the heterogenous oxidation mechanism in detail in this paper. However, thanks to the reviewers' comment we realized that we should elaborate in more detail on possible underlying processes. As we could observe a clear dependency of the transformation kinetics on the presence of oxygen, we name/discuss in the revised manuscript three different plausible processes well described in literature, how oxygen interacts with MnO_2 and could potentially influence (increase) the DTPMP oxidation by MnO_2 . The text now reads:

Lines 150 to 162:

The rate enhancing role of oxygen may be related to multiple processes involving different redox states of manganese as reported in literature. Under oxic conditions, Mn^{2+} (formed by the reduction of Mn^{IV} ^{26,30}) is known to catalyze DTPMP (and ATMP, EDTMP) oxidation by O_2 (Mn^{2+}/O_2) in homogeneous solution^{28,34}. However, the much slower reaction kinetics of DTPMP in homogeneous solution (1 mM Mn^{2+} and O_2) compared to the heterogeneous systems found in this study clearly show that the strongly enhancing role of O_2 in MnO_2 experiments must be due to processes other than mere Mn^{2+}/O_2 interaction, probably involving the formation of Mn^{III} on the mineral surface.

Manganese minerals with elevated Mn^{III} content appear to be more reactive oxidants^{30,35,36}. In heterogeneous systems containing Mn^{2+} and MnO_2 , Mn^{III} can be formed by comproportionation and be associated with the mineral surface or reside in solution³⁵. Furthermore, MnO_2 can catalyze the oxidation of Mn^{II} by O_2 to Mn^{III} ³⁷. Finally, MnO_2 itself may act as a direct oxidant but also as a catalyst in connection with O_2 ³⁸.

9. Line 324: use of the buffer: does the buffer interact with MnO_2 as well? Is the pH change after the reaction monitored?

Response:

We would like to thank the reviewer for the comment. We asked this question ourselves and therefore investigated the influence of MES within our work from last year (Röhnelt et al. 2023)¹⁶, where we investigated IDMP transformation on MnO_2 . We conducted the same experiment twice, keeping the pH constant by i) buffering with MES or ii) titrating, and did not observe any difference in IDMP transformation. (The data is shown in the Supporting Information page 3 of that manuscript.) As higher APPs such as DTPMP are sorbing much stronger to MnO_2 and form much stronger complexes with Mn^{2+} ²⁴, we consider it plausible that MES does not have any effect on DTPMP transformation by either MnO_2 or Mn^{2+}/O_2 .

We thank the reviewer for the suggestion concerning the pH stability. We now included Figure S1 in the Supporting Information showing the pH over time in several experiments and controls. The pH remained stable with a deviation of $< \pm 0.4$ pH-units for each individual experimental condition. The first paragraph in “DTPMP transformation by manganese” of “Results & Discussion” now reads:

Lines 123 to 128:

The experiments were carried out in purified water (buffered at pH 6) as well as in sterile-filtered wastewater (pH 8) as matrix. The pH values were monitored and are depicted in Fig. S1.

Figure S 1: pH values over time in four different experiments (Exp.) including controls (c1-c3, respectively).
a): Exp.: 1.0 g/L MnO₂ ox in MES; b): Exp.: 1 mM MnCl₂ ox in MES, c1: no MnCl₂; c): Exp.: 1.0 g/L MnO₂ ox in wastewater, c1: pure wastewater, c2: no MnO₂, c3: no DTPMP; d): Exp.: 1 mM MnCl₂ ox in wastewater, c1: no DTPMP.

Reviewer #2 (Remarks to the Author):

General remarks:

The authors of this manuscript were members of a team that published a paper in Water Research (ref 14) in October 2024 suggesting that municipal wastewater treatment plants in Europe are important, but under-recognized sources of glyphosate. They also proposed that aminopolyphosphonates, mainly from laundry detergents are transformed to produce glyphosphate. In this manuscript, the authors propose that manganese in wastewater is responsible for the conversion of the aminopolyphosphonates diethylenetriamine penta(methylene phosphonate) (DTPMP) into glyphosate through a reaction pathway that has a relatively low yield (i.e., <0.2%). Overall, I found the hypothesis of the authors to be interesting and the topic of the research to have substantial implications for efforts to control glyphosate pollution. Recognizing that this is an initial communication, it seems reasonable that some of the details needed to fully characterize these reactions might not be available. Nonetheless, I remain unconvinced that the authors have proven that this reaction is the source of the glyphosate. Given the potential significance of the findings of this study, I believe that a more thorough analysis of the processes and a better justification of the experimental conditions is needed prior to publication.

General response:

We thank the reviewer for the positive assessment and insightful criticism of our work. To address the environmental relevance of our study, we performed additional experiments in sterile-filtered wastewater. The results of these experiments demonstrate that glyphosate and AMPA can be formed from DTPMP via manganese-driven oxidation under near-environmental conditions. Furthermore, the evaluation by reviewer #2 highlighted potential issues with the original manuscript. To address this, we have rephrased certain sections of the text in the revised version, emphasising that the study's conclusion is not the identification of the source of glyphosate in surface waters. Instead, the aim is to present a novel hypothesis regarding a potential source.

Please find our response to detailed questions below.

My specific concerns about the manuscript are detailed below.

1. If I understand the hypothesis correctly, the authors set out to determine if the reaction between DTPMP and Mn takes place between the location where it is first used (e.g., in

laundry washing machines) and the discharge of a conventional wastewater treatment plant. The experiments described in the manuscript do not give me confidence that the reaction has been studied under environmentally relevant conditions as the authors claim. My first concern is that most chelating reagents like DTPMP will exist as complexes with metals. Have the authors proven that glyphosate is produced when the DTPMP is complexed by Fe(III), Ca(II) or other metals that are present in sewage?

Response:

We thank the reviewer for that detailed feedback. Since the comment touches on several issues, we will address them one by one.

a) Research questions and aim of this work

We would like to thank the reviewer for her/his constructive feedback. Aminopolyphosphonates are often reported to be rather persistent^{12,25-27}. Schwientek et al. (2024)²⁸ raised the question whether APPS are a source of glyphosate, which would then have to form in WWTPs. However, AAPs are reported not to be well biodegradable²⁹, which raises the question if other than biotic pathways may be relevant. With our previous experience on phosphonate reaction with manganese oxides¹⁶, we investigated this abiotic reaction pathway to learn, if glyphosate may – at all – form from DTPMP. First, aqueous model solutions were investigated at circumneutral pH and the relevance of oxygen was considered as would be the case under oxic/anoxic conditions in WWTPs and in sediment.

The knowledge on glyphosate formation from DTPMP by manganese-driven oxidation being chemically feasible presents great indicators for practical research. Mn oxides are important oxidants in the environment, Mn is ubiquitous in the environment and – in quite high concentrations – in wastewater sludge².

Thanks to this comment we realized that we needed to redefine the scope and aim of this study – a proof-of-concept work that shows that glyphosate formation from DTPMP is chemically feasible under technically and environmentally relevant reaction conditions (circumneutral pH, technically/environmentally relevant oxidant).

Hence, we revised some text parts, they now read:

Lines 32 to 39:

The ubiquitous presence of manganese in natural and engineered systems underscores the potential importance of Mn-driven DTPMP transformation as a previously overlooked source of glyphosate in aquatic systems. While further research is needed to evaluate the factors controlling the product spectrum and glyphosate yields from DTPMP transformation in technical and natural settings, our results challenge the current

paradigm that herbicide application is the sole source of environmental glyphosate contamination. Consequently, current strategies and approaches to protect water resources from glyphosate contamination need to be reconsidered and expanded.

Lines 317 to 324:

Our study demonstrates for the first time that manganese potentially plays a key role in converting the widely used complexing agent DTPMP in a multi-step reaction to the herbicide glyphosate. The reaction proceeds at circumneutral pH at MnO_2 minerals both in the absence and presence of dissolved oxygen but also in homogeneous solution in the presence of $\text{Mn}^{2+}/\text{O}_2$, even in wastewater. Under all conditions studied, AMPA and glyphosate were transformation products, AMPA up to 27.1 mol-% and glyphosate up to 0.42 mol-%.

Lines 355 to 358:

Overall, our work offers a scientific basis to rationalize recent and unexpected findings of elevated glyphosate concentrations in European WWTP effluents^{12,15} and suggests that manganese may play a crucial role in this phenomenon, potentially serving as a key factor in understanding the underlying mechanisms.

b) Environmental relevant conditions

Next, we would like to focus on the question concerning environmentally relevant conditions and especially metal cations present in solution. We thank the reviewer for this important comment and agree that in technical/natural systems high fractions of APPs are complexed, mainly with calcium and magnesium³⁰. In order to account for this in the revised version of the manuscript, we collected wastewater (pH 8) just after the mechanical cleaning in the wastewater treatment plant in Tübingen, southwest Germany. We conducted new experiments with this undiluted matrix with both MnO_2 (i) and $\text{Mn}^{2+}/\text{O}_2$ (ii) as the oxidant/catalyst. The wastewater contained (inter alia) 72 mg/L (1.8 mM) Ca^{2+} and 13 mg/L Mg^{2+} (0.5 mM), as well as 100 mg/L SO_4^{2-} (1.0 mM) and 11.6 mg/L DOC.

In this wastewater matrix we observed glyphosate and AMPA formation. Interestingly, the experiment with wastewater and 1 mM Mn^{2+} under oxic conditions led to the highest glyphosate yield of all experiments (0.42 mol%). We are confident that the results of the new experiments improve the manuscript and strengthen its relevance.

The new experimental results are added as a new subchapter “Formation of glyphosate and AMPA from DTPMP in wastewater matrix” as follows:

Lines 272 to 307:

To address the environmental relevance of the observations in pure water, experiments containing 1 g/L MnO₂ and 1 mM Mn²⁺ were conducted in wastewater (pH 8, sterile filtrated; see Materials & Methods for details on the wastewater sample). In control experiments with unspiked wastewater with and without MnO₂ or Mn²⁺, negligible glyphosate and AMPA concentrations were occasionally detected (see Fig. S6). Dissolved manganese in the wastewater sample was below the detection limit (<0.04 mg/L).

DTPMP transformation kinetics with 1.0 g/L MnO₂ in the wastewater matrix were slower than in MES buffer at pH 6 (see Fig. S7 b) in line with a lower oxidation potential of manganese oxides at higher pH^{47,48}, as well as increased electrostatic repulsion between DTPMP and the mineral surface (point of zero charge of 5.6)^{30,49}. Furthermore, differences between the experiments with MES buffer and wastewater are likely due to the complex wastewater-matrix containing organics (such as other complexing agents) and cations (e.g., calcium), which were reported to influence APP transformation and sorption^{18,50}. Glyphosate and AMPA yields were up to 0.06 mol-% (glyphosate) and 10.3 mol-% (AMPA) after 168 hours. However, in wastewater spiked with 1 mM Mn²⁺, DTPMP transformation kinetics were faster and had higher glyphosate and AMPA yields compared to MES-buffered pure water at pH 6 (see Fig. 5.) In wastewater spiked with 1 mM Mn²⁺, the highest glyphosate yield (0.42 mol-%) of all experiments conducted in this study was observed (see Table 2), albeit only after 240 h. Possibly, the reaction kinetics are faster at pH 8 due to stronger complex formation of Mn^{II} and DTPMP, such as shown for ATMP²⁸. At higher pH, less protons compete with metal ions present in solution and DTPMP is more negatively charged⁵¹. Furthermore, Mn^{III} complexes might play a role at higher pH. While the stability of Mn^{III} complexes varies depending on the ligand, certain ligands including desferrioxamine B show higher stability of Mn^{III} complexes at pH values between 7-11⁵².

Fig. 5 DTPMP (black), glyphosate (red) and AMPA (blue) concentrations during oxidation of 1 mM DTPMP by 1 mM Mn²⁺ in the experiments containing **a)** pure 20 mM MES buffer (pH 6) and **b)** wastewater (pH 8) as matrices. DTPMP was quantified using IC-PAD, glyphosate and AMPA were quantified using LC-QQQ. Error bars represent absolute errors between experimental duplicates.

Table 2 Maximum total AMPA and glyphosate yields given in mol-% of the initial quantified DTPMP concentration. Errors for the latter represent absolute errors between duplicates. MES denotes experiments in aqueous 20 mM MES buffer at pH 6, while WW stands for sterile-filtered wastewater at pH 8. The timepoint denotes the time of maximum observed AMPA resp. glyphosate formation.

c(MnO₂) in g/L	c(Mn²⁺) in mM	O₂	Matrix	AMPA_{max} in mol-%	Timepoint in h	Gly_{max} in mol-%	Timepoint in h
1.0	-	oxic	MES	10.1 ± 0.2	3.5	0.16 ± 0.02	1.0
0.1	-	oxic	MES	4.5 ± 0.2	3.0	0.16 ± 0.01	28
1.0	-	anoxic	MES	5.0 ± 1.0	6.0	0.06 ± 0.00	24
0.1	-	anoxic	MES	10.1 ± 0.5	24	0.03 ± 0.00	24
1.0	-	oxic	WW	10.3 ± 0.3	168	0.06 ± 0.01	168
-	1.0	oxic	MES	6.7 ± 0.7	185	0.07 ± 0.00	185
-	1.0	oxic	WW	27.1 ± 0.5	240	0.42 ± 0.01	240

The wastewater characterization can be found in the Materials & Methods part and in the SI.

Materials & Methods

Lines 434 to 442:

The wastewater was sampled at 10 am on September 9, 2024 from the municipal wastewater treatment plant in Lustnau (Tübingen, SW Germany), at the outflow of the mechanical treatment. The wastewater was then filtered with different filter systems: I) coffee filter (Melitta, Minden, Germany), II) folded filters 595 ½ (Whatman Int. Ltd, Buckinghamshire, UK), III) glass fibre round filters GF 55 (Schleicher & Schuell, Dassel, Germany) and finally IV) sterile S-PAK 0.22 µm filters (Merck, Darmstadt, Germany). This filtered wastewater was used undiluted as matrix for the experiments. The initial pH of the wastewater was 8. The changes in pH development over time are shown in Fig. S1. Detailed information regarding the composition of the wastewater are provided below.

Lines 560 to 576:

Wastewater Characterization

The wastewater sample taken from the WWTP in Lustnau (Tübingen, Germany) on September 10, 2024, had a pH value of 7.94. The dissolved organic carbon (DOC) measured as non-purgeable organic carbon (NPOC), was determined to be 11.6 mg/L. Table 7 summarizes the results of the wastewater sample characterization using IC with conductivity detection and MP-AES (for analytical methods see the Supporting Information). Table 8 contains additional information on the 24h-mixed sample monitored by the wastewater treatment plant.

Table 7 Anions and cations quantified in the wastewater sample using IC with conductivity detection, except for Mn and Fe, which were quantified using microwave-plasma atomic emission spectroscopy (MP-AES), and therefore are not assigned cationic charges.

Anion	c in mg/L	Cation	c in mg/L
F ⁻	0.23	Na ⁺	27.9
Cl ⁻	39.1	NH ₄ ⁺	13.0
NO ₂ ⁻	0.7	K ⁺	7.8
Br ⁻	0.04	Mg ²⁺	13.3
NO ₃ ⁻	7.1	Ca ²⁺	71.3
PO ₄ ³⁻	3.3	Mn	<0.04
SO ₄ ²⁻	99.4	Fe	<0.02

Table 8 Wastewater parameters recorded for a 24-hour mixed sample in the WWTP Lustnau on September 10, 2024. COD stands for “chemical oxygen demand”.

Parameter	Value
COD	195 mg/L
P total	2.27 mg/L
Conductivity	99.36 mS/cm
Temperature	18.8 °C

2. Similarly, the authors chose to study extremely high concentrations of a crystalline manganese oxide. The form of the manganese oxide is known to play a very significant role in the transformation of organic compounds. I was disappointed in the supporting information, which did not give data on the polymorph of Mn oxide. Moreover, I do not understand the justification for assuming the presence of 100 or 1000 mg/L of pure Mn oxides in wastewater. The suspended solids in raw sewage or activated sludge mainly consist of organic matter with relatively low concentrations of minerals, like silica and iron oxides. Even under the unlikely situation in which sewage were to contain high concentrations of manganese oxides, it is likely that the surfaces would be covered with protein, biopolymers, surfactants and other substances that could block or slow the interactions of DTPMP with the surface. Under some conditions, sulfide might also form other Mn-containing minerals (e.g., MnS). Can the authors better justify the concentrations of Mn oxides used in these experiments? Do these reactions take place in the presence of other substances present in sewage?

Response:

Again, we thank the reviewer for those important suggestions. As before, we will refer to different aspects of that comment separately:

a) MnO₂ polymorphism

Referring to the question about the MnO₂ polymorph: Indeed, the MnO₂ crystal structure has a great influence on the reactivity. However, as this is a proof-of-concept study, we do not regard the detailed discussion of the used MnO₂ as relevant. Still, we agree with the reviewer that it is important to provide data on the mineral for future research. Therefore, we recorded XR diffractograms of the MnO₂ used in this study (Figure S11). We now added a sentence describing the XR diffractograms:

Lines 553 to 558:

The point of zero charge (pH_{PZC}) of the manganese dioxide was determined at pH 5.6 ± 0.1 using zeta potential measurements. The X-ray diffractogram (see Fig. S11) showed a mostly amorphous structure, interspersed with some crystalline domains (pyrolusite, akhtensite).

As Mn oxides in natural environments are primarily produced biogenically^{7,31}, they mostly exhibit an amorphous structure^{1,17,32}. Thus, we regard the used mineral as a good representative for the MnO₂ family found in natural environments.

b) Mn oxide concentration

As reviewer #1 asked the same question concerning the MnO_2 concentration, we copied the answer from above:

Our work aims to provide experimental evidence for the formation of glyphosate from DTPMP in the presence of manganese. As such, it is designed as a proof-of-concept study that sets the stage for follow-up studies addressing more specific objectives, such as the extent to which such processes can explain elevated glyphosate concentrations observed in surface waters.

The mere comparison of total Mn or MnO_x contents in natural or technical environments with the MnO_2 mineral concentrations used in this study to judge on its environmental relevance is more challenging than it seems. This is due to the importance of MnO_x surface areas for the mineral's reactivity, as reactions occur at the surface. Mn oxides often occur as fine-grained particles or coatings in the environment¹, which increases the available surface and therefore the reactivity. In our case, we used pure Mn oxide. Nevertheless, we compiled total Mn contents in natural and technical environments to contextualize our concentrations:

The range of Mn concentration in wastewater sludge is 600 – 1,500 mg/kg resp. 3,300 – 3,900 dry mass according to the German environment agency² resp. Milik et al. (2017)³. The moisture content in sewage sludge is typically $\geq 90\%$, depending on the type of sludge³. Upon the assumption of 95 % moisture content, Mn concentrations regarding the wet mass are in the range of 30 to 195 mg/kg. Hence our lower MnO_2 concentration (0.1 g/L) matches the range of total Mn in wet wastewater sludge. The total Mn content in topsoils typically ranges between 20 and 3000 mg/kg⁴ with typical levels around 600 mg/kg⁵. The porosity of hyporheic zone sediments is usually in the range of 0.3 to 0.4, thus by assuming a porosity of 0.35 we arrive at about 400 mg/kg wet mass. The equilibrium between MnO_x minerals and Mn^{2+} in solution depends on the pH, redox conditions (Mn^{III} and Mn^{IV} oxides are generally more stable at higher pH values and under oxidizing conditions⁶ and in the presence of Mn^{2+} oxidizing micro-organisms⁷).

c) Presence of other substances

We thank the reviewer for this suggestion. We conducted new experiments in a wastewater matrix. The wastewater contained a DOC concentration of 11.5 mg/L, cation and anion concentrations can be found in Materials & Methods, "Wastewater Characterization". Even in this matrix, DTPMP was transformed, while glyphosate and AMPA formed. The experiment containing MnO_2 proceeded slower in wastewater (pH 8) than its counterpart in ultrapure water with MES buffer (pH 6), while the experiment containing Mn^{2+} proceeded faster in wastewater. The new experimental results are added as a new subchapter "Formation of glyphosate and AMPA from DTPMP in wastewater matrix" as follows:

Lines 272 to 295:

To address the environmental relevance of the observations in pure water, experiments containing 1 g/L MnO₂ and 1 mM Mn²⁺ were conducted in wastewater (pH 8, sterile filtrated; see Materials & Methods for details on the wastewater sample). In control experiments with unspiked wastewater with and without MnO₂ or Mn²⁺, negligible glyphosate and AMPA concentrations were occasionally detected (see Fig. S6). Dissolved manganese in the wastewater sample was below the detection limit (<0.04 mg/L).

DTPMP transformation kinetics with 1.0 g/L MnO₂ in the wastewater matrix were slower than in MES buffer at pH 6 (see Fig. S7 b) in line with a lower oxidation potential of manganese oxides at higher pH^{47,48}, as well as increased electrostatic repulsion between DTPMP and the mineral surface (point of zero charge of 5.6)^{30,49}. Furthermore, differences between the experiments with MES buffer and wastewater are likely due to the complex wastewater-matrix containing organics (such as other complexing agents) and cations (e.g., calcium), which were reported to influence APP transformation and sorption^{18,50}. Glyphosate and AMPA yields were up to 0.06 mol-% (glyphosate) and 10.3 mol-% (AMPA) after 168 hours. However, in wastewater spiked with 1 mM Mn²⁺, DTPMP transformation kinetics were faster and had higher glyphosate and AMPA yields compared to MES-buffered pure water at pH 6 (see Fig. 5.) In wastewater spiked with 1 mM Mn²⁺, the highest glyphosate yield (0.42 mol-%) of all experiments conducted in this study was observed (see Table 2), albeit only after 240 h. Possibly, the reaction kinetics are faster at pH 8 due to stronger complex formation of Mn^{II} and DTPMP, such as shown for ATMP²⁸. At higher pH, less protons compete with metal ions present in solution and DTPMP is more negatively charged⁵¹. Furthermore, Mn^{III} complexes might play a role at higher pH. While the stability of Mn^{III} complexes varies depending on the ligand, certain ligands including desferrioxamine B show higher stability of Mn^{III} complexes at pH values between 7-11⁵².

3. The experiments were conducted at pH 6 using MES as a buffer. Oxidation reactions involving Mn-oxides tend to exhibit a strong pH dependence. Were these reactions replicated at higher pH values (e.g., pH 7-8 are typical of sewage and treated wastewater)?

Response:

We thank the reviewer for this important suggestion. We conducted new experiments with wastewater at pH 8. The new experimental results are added as a new subchapter "Formation of glyphosate and AMPA from DTPMP in wastewater matrix" in lines 272 to 295,

Figure 5 and Figure S7. As expected, DTPMP oxidation on MnO₂ was slower in this matrix at pH 8, as the oxidation capacity of MnO₂ depends on the pH. Still, AMPA and glyphosate formation was monitored. The DTPMP oxidation by Mn²⁺/O₂ on the contrary was faster than compared to pH 6 and solely MES buffer.

We wrote a new subchapter “Formation of glyphosate and AMPA from DTPMP in wastewater matrix”, which is inserted in our answer above.

4. The reaction between MnO₂ and DTPMP appears to have been studied by other authors. The formation of glyphosate as a minor transformation product seems to be thermodynamically possible and the fact that it is a minor pathway could help explain the fact that this reaction has not been considered by others. However, the lack of a quantitative treatment of the observed formation of glyphosate leaves the reader wondering whether the explanation is valid. If data are available on DTPMP concentrations in sewage and wastewater treatment plants, it should be possible to convert the observed losses of the parent compound into glyphosate concentrations formed in the process by using the yield measurements. Have the authors made these kinds of calculations? If so, can they explain the observations reported in the Water Research paper?

Response:

We thank the reviewer for the comment. We will answer separately:

a) Other research on DTPMP oxidation by MnO₂

To our knowledge, there is no published work investigating the oxidation of DTPMP by MnO₂. However, there is research investigating ATMP, EDTMP and DTPMP transformation on Mn²⁺/O₂^{10,33}), and ATMP transformation on MnOOH^{8,9,33}. There is our work from previous year investigating IDMP oxidation by MnO₂¹⁶ and multiple studies investigating glyphosate and/or AMPA oxidation by MnO_x³⁴⁻³⁷. Therefore, no comparison to earlier work is possible and no changes to the text were made.

b) Quantitative assessment of big scale glyphosate formation

As mentioned in the general answer to reviewer #2, we realized that the scope of the work could be misunderstood in the original version of the manuscript. The aim of this manuscript is to present a “proof of concept” study, in which we show, that glyphosate can form during DTPMP oxidation by an environmentally relevant oxidant and under environmentally relevant conditions (pH 6 and 8, see new experiments using wastewater). We provide information and data from controlled laboratory experiments and point out manganese (either (i) Mn²⁺ in the

presence of O₂ or (ii) MnO₂) as an important oxidant found in the environment and in WWTPs, potentially playing a role in DTPMP transformation in natural and technical systems.

We now clarified in the manuscript that although the study provides experimental evidence that glyphosate forms from manganese-driven DTPMP transformation under environmentally relevant conditions, it cannot be used to quantitatively assess de novo glyphosate formation and inputs to European surface waters.

Lines 32 to 39:

The ubiquitous presence of manganese in natural and engineered systems underscores the potential importance of Mn-driven DTPMP transformation as a previously overlooked source of glyphosate in aquatic systems. While further research is needed to evaluate the factors controlling the product spectrum and glyphosate yields from DTPMP transformation in technical and natural settings, our results challenge the current paradigm that herbicide application is the sole source of environmental glyphosate contamination. Consequently, current strategies and approaches to protect water resources from glyphosate contamination need to be reconsidered and expanded.

Lines 339 to 345:

Overall, this study for the first time provides experimental evidence for conversion of a widely used non-toxic commodity compound into a highly debated pesticide³⁸⁻⁴¹ under environmentally relevant reaction conditions. While we could demonstrate that the reaction is chemically feasible in the laboratory, future research should elucidate in detail how environmental conditions affect glyphosate formation from DTPMP and related APPs, the formation and identification of key intermediates and field studies including yields in wastewater treatment plants.

Lines 355 to 358:

Overall, our work offers a scientific basis to rationalize recent and unexpected findings of elevated glyphosate concentrations in European WWTP effluents^{28,42} and suggests that manganese may play a crucial role in this phenomenon, potentially serving as a key factor in understanding the underlying mechanisms.

A comprehensive investigation into the processes occurring within wastewater treatment plants (WWTPs) or sewer systems, and to which extent these can account for the glyphosate concentrations detected in surface waters, constitutes a separate research endeavor, which will be addressed in multiple follow-up studies. We consider the proposed calculations based on our yield data to be speculative. Our experiments are primarily designed to address

fundamental research questions. The complexity of processes in WWTPs far exceeds the scope of our current study. Therefore, we are hesitant to extrapolate quantitative estimates from our yields. Robust calculations require more research to investigate actual processes and glyphosate yields across different WWTPs, potentially using isotope-labeled DTPMP. Additionally, accurate usage volumes of DTPMP (and EDTMP, as it could also potentially form glyphosate) remain unknown.

Consequently, we conclude that there is insufficient data on both glyphosate yields and DTPMP (and EDTMP) concentrations to perform these calculations with confidence, and we do not consider this aspect within our current research scope.

5. Consideration of the fate of DTPMP in wastewater without discussion of the possible importance of biotransformation of the compound seems incomplete. Is this compound transformed in sewage or in sewage treatment plants? If so, could biotransformation explain the observed formation of glyphosate?

Response:

We thank the reviewer for that comment. As another reviewer had the same question, we copy our answers from above:

a) Biodegradation of aminopolyphosphonates

Knowledge on the biotransformation of aminopolyphosphonates (APPs) like DTPMP is still scarce. Few studies investigated APP biotransformation, but the presence of manganese in all bacterial growth media and the observation of APP transformation in abiotic controls (DTPMP in the pure medium without bacteria) or the lack of sufficient negative controls leaves serious doubt that the microbes are using the DTPMP as phosphorus source. It seems conceivable that the microbes use PO_4^{3-} -P or AMPA-P resulting from abiotic Mn-driven transformation of DTPMP to facilitate growth, due to the presence of Mn^{2+} and O_2 . The abiotic transformation of aminotris(methylenephosphonate) (ATMP) by Mn^{2+}/O_2 was already investigated in detail in previous studies⁸⁻¹⁰. It could be shown that even if the Mn^{2+} :ATMP ratio is as low as 1:100, ATMP is completely transformed after approximately 30 hours. Negative controls from i) Steber & Wierich (1987)¹¹, ii) Schowanek & Verstraete (1990)¹² and iii) Drzyzga & Lipok (2017)¹¹ showed chemical ATMP (i and iii) and GBMP (iii) and DTPMP (ii and iii) transformation in non-inoculated growth media in the dark, while Riedel et al. (2023 and 2024a and 2024b)¹³⁻¹⁵ miss cell free controls of DTPMP in the used bacterial growth medium. This raises significant questions regarding the occurrence of the actual extent of DTPMP biotransformation in these studies.

The formation of glyphosate from DTPMP biotransformation has never been reported anywhere until now.

However, thanks to the reviewers' comments, we realized there was a gap in our manuscript. In the revised version of the manuscript, we amended the conclusion with a small paragraph on biotic APP degradation and the role of manganese in the bacterial growth medium. It now reads:

Lines 346 to 354:

In addition, our findings may also provide new clues for revisiting APP biotransformation studies. All bacterial growth media used in published APP biotransformation studies contain dissolved manganese^{53,57-61}. Thus, manganese-driven oxidation may occur in parallel with or instead of biotransformation of DTPMP and other APPs. Martin et al (2022)¹⁰ showed that even at a molar ratio of 1:100 (Mn:ATMP), ATMP was completely degraded within 30 hours. Thus, it is conceivable that in biotransformation studies under oxic conditions in the presence of dissolved Mn²⁺, microorganisms may utilize chemical transformation products of APPs, such as phosphate, IDMP, or AMPA, as a phosphorus source rather than or in addition to directly metabolizing the target APPs.

6. The discussion of reaction kinetics needs considerable improvement. For example, I found the description of pseudo-zero order kinetics to be quite confusing. Yes, one might observe zero-order kinetics for some catalytic reactions, but drawing lines based on one or 4-5 data points (i.e., those where the parent compound is present at concentrations above the detection limit and then calculating r² values and rate constants to four significant figures and attempting to interpret mechanisms is an over-reach.

Response:

We thank the reviewer for the comment. We think we might have unfortunately phrased the paragraph about DTPMP transformation in a way that we arrived at some kind of misunderstanding. We do not want to interpret mechanisms based on the pseudo-0th-order reaction rates. The reason to derive reaction rates in the first place was merely to compare the different experiments with respect to kinetics. As the reaction mechanism is quite complex, discovering the "actual" reaction order and/or deriving a model suiting all experiments is challenging. The pseudo-0th order kinetics resulted in satisfactory fits, while the experimental data did not conform to either first or second order kinetic models. Therefore, we have chosen a linear regression (pseudo-0th order) as an approximation to compare the reaction speeds.

Thanks to this comment, we adjusted the text to make clear that the fits are an approximation to allow relative comparison:

Lines 134 to 139:

To evaluate the DTPMP transformation kinetics, pseudo 0th-order rate constants were determined as no higher reaction order adequately described the kinetics across all four MnO₂ experiments. These constants were derived by linear regression considering the time intervals described in Table 1. This approach allowed for a comparative kinetic analysis of the four MnO₂ experiments.

Regarding the fits, we agree with the reviewer that we might have overstated their statistical quality. Thus, we have reduced the number of significant digits for the rate constants and R² values. This shall provide more transparency for the numbers derived. To be more transparent with respect to the data point used for the fits, we also amended Table 1 by the time interval included in the linear regression.

Table 1 Pseudo-0th order reaction rate constants (k) and those reaction rate constants normalized to the surface area (k_{norm}) for DTPMP transformation in the four experiments with MnO₂ in MES buffer. The standard errors of the linear regression are given as ±x. R² is the regression coefficient of the linear regression from the start of the experiment until complete DTPMP transformation (24 h excluded for 0.1 g/L anoxic due to low data point density). The linear section indicates the time interval/timepoints included in the linear regression.

c(MnO₂)	1.0 g/L	0.1 g/L	1.0 g/L	0.1 g/L
O₂	oxic	oxic	anoxic	anoxic
k in μM/h	2729 ± 800	650 ± 106	946 ± 26	133 ± 3
k_{norm} in μmol/(m²*h)	42 ± 12	101 ± 16	14.7 ± 0.4	20.6 ± 0.5
R²	0.842	0.859	0.996	0.988
Linear section in h	0-0.35	0-1.5	0-0.67	0-6.0

By deriving the pseudo-zero order reaction rate constants, we noticed that the anoxic reactions could be described well by a linear fit, while the oxic experiments could not. This was mentioned, as it could be a hint for some other mechanism/other parallel reactions taking place. A possible explanation would be e.g., direct oxidation (oxic & anoxic) and catalytic activity of the MnO₂ in presence of O₂ (oxic experiments), as oxic experiments were

much faster. Subsuming, we merely used pseudo-0th order reaction rates to relatively compare the four experiments with each other, but not to make detailed mechanistic interpretations from linear regressions. Additionally, the derivation of rate constants enabled us to normalize them to the available surface area of the mineral. This is now included in the discussion and Table 1 in the revised version of the manuscript. It now reads:

Lines 146 to 157:

Normalization of the reaction rate constants to the specific surface area (k_{norm}) provided further insights into the role of oxygen and available surface area. k_{norm} -values showed that the oxic experiments were faster compared to their anoxic counterparts.

The rate enhancing role of oxygen may be related to multiple processes involving different redox states of manganese as reported in literature. Under oxic conditions, Mn^{2+} (formed by the reduction of Mn^{IV} ^{26,30}) is known to catalyze DTPMP (and ATMP, EDTMP) oxidation by O_2 ($\text{Mn}^{2+}/\text{O}_2$) in homogeneous solution^{28,34}. However, the much slower reaction kinetics of DTPMP in homogeneous solution (1 mM Mn^{2+} and O_2) compared to the heterogeneous systems found in this study clearly show that the strongly enhancing role of O_2 in MnO_2 experiments must be due to processes other than mere $\text{Mn}^{2+}/\text{O}_2$ interaction, probably involving the formation of Mn^{III} on the mineral surface.

7. The authors state that Mn^{2+} also oxidizes DTPMP but they did only included results from a single experiment conducted with 1 mM of DTPMP and 1 mM Mn^{2+} . In this experiment products were only observed after about 5 days. I struggle to relate this very unrealistic experiment and the timeframes needed to see the formation of glyphosate to the overall hypothesis of the authors. Stating that Mn^{2+} and oxygen can produce glyphosate under environmentally relevant conditions, as stated in the abstract, seems like another overreach.

Response:

We thank the reviewer for that comment. Indeed, we only showed a single DTPMP + $\text{Mn}^{2+}/\text{O}_2$ experiment in the first version of the manuscript.

Thanks to the reviewers' comments we repeated the experiment and monitored it for a longer time span, i.e, 180 hours. The DTPMP data is included in the Supporting Information in Figure S3, while DTPMP, glyphosate and AMPA concentrations over time are depicted in Figure 5 a)

in the main text. The corresponding paragraph describing glyphosate and AMPA formation in this experiment now reads:

Lines 247 to 253

To elucidate the significance of heterogeneous (MnO_2) and homogeneous ($\text{Mn}^{2+}/\text{O}_2$) oxidation reactions on product formation, an experiment with 1 mM DTPMP and 1 mM dissolved Mn^{2+} (MnCl_2) was conducted under oxic conditions (buffered at pH 6). Neither glyphosate nor AMPA formation was observed within the first 24 hours (see Fig. 5 a). After 137 hours (~5.5 days), however, 6.3 ± 0.2 mol-% AMPA and 0.06 ± 0.01 mol-% glyphosate were quantified. AMPA and glyphosate concentrations stayed almost constant until 185 h yielding 6.8 ± 0.7 mol-% (AMPA) and 0.07 ± 0.00 mol-% (glyphosate).

Figure 5: DTPMP (black), glyphosate (red) and AMPA (blue) concentrations during oxidation of 1 mM DTPMP by 1 mM Mn^{2+} in the experiments containing **a)** pure 20 mM MES buffer (pH 6) and **b)** wastewater (pH 8) as matrices. DTPMP was quantified using IC-PAD, glyphosate and AMPA were quantified using LC-QQQ. Error bars represent absolute errors between experimental duplicates.

Further, we conducted a new experiment investigating DTPMP oxidation by $\text{Mn}^{2+}/\text{O}_2$ in wastewater (Figure 5 b). This new experiment shows the highest glyphosate yield of all experiments and thus clearly underlines the environmental relevance of this process. The experimental results of both experiments (in MES and in wastewater matrix), together with an experiment containing 1 g/L MnO_2 in wastewater are now added to Table 2.

Table 2 Maximum total AMPA and glyphosate yields given in mol-% of the initial quantified DTPMP concentration. Errors for the latter represent absolute errors between duplicates. MES denotes experiments in aqueous 20 mM MES buffer at pH 6, while WW stands for sterile-filtered wastewater at pH 8. The timepoint denotes the time of maximum observed AMPA resp. glyphosate formation.

c(MnO₂) in g/L	c(Mn²⁺) in mM	O₂	Matrix	AMPA_{max} in mol-%	Timepoint in h	Gly_{max} in mol-%	Timepoint in h
1.0	-	oxic	MES	10.1 ± 0.2	3.5	0.16 ± 0.02	1.0
0.1	-	oxic	MES	4.5 ± 0.2	3.0	0.16 ± 0.01	28
1.0	-	anoxic	MES	5.0 ± 1.0	6.0	0.06 ± 0.00	24
0.1	-	anoxic	MES	10.1 ± 0.5	24	0.03 ± 0.00	24
1.0	-	oxic	WW	10.3 ± 0.3	168	0.06 ± 0.01	168
-	1.0	oxic	MES	6.7 ± 0.7	185	0.07 ± 0.00	185
-	1.0	oxic	WW	27.1 ± 0.5	240	0.42 ± 0.01	240

We do not regard the observed time frames as a problem, but rather as another argument for the process' environmental relevance, as glyphosate and AMPA are stable for days after complete DTPMP transformation in the reaction solutions and suspensions.

Reviewer #3 (Remarks to the Author):

General remarks:

The study shows for the first time that glyphosate can be produced in trace amounts by the reaction of an aminophosphonate with manganese oxides under neutral pH conditions. This is a remarkable result in the field of water pollution, as the origin of glyphosate in the environment has recently been the subject of debate in Europe. The context is well described and supported by appropriate, up-to-date references. This study clearly shows that glyphosate can be produced by the transformation of a widely used non-toxic organic compound, and not just by its use in agriculture.

Glyphosate is not the main transformation product, which explains why it has been overlooked until now. Thus, specific analytical methods have been used in this study for its analysis at trace level. Results also show that concentrations of manganese oxide and dissolved oxygen strongly influence the yield of glyphosate formation and that glyphosate can also be formed in presence of dissolved manganese ions and oxygen for long reaction time. Interesting result also is the stability of glyphosate in presence of an excess of manganese oxide.

A reaction scheme is proposed based on literature data, which will need to be revised slightly (see remark below). The data are of high quality and obtained with the appropriate protocols and analytical methods. Protocols and methods are generally well described.

The authors suggest that this reaction can occur in wastewater treatment plants because aminophosphonates are used as complexing agents with detergents and manganese oxides are present in sewage sludge. However, the authors do not demonstrate that this reaction actually occurs in wastewater treatment plants. The experiments were carried out with a high concentration of aminophosphonate (1 mM) in pure aqueous suspensions of manganese oxides. The lower concentration of aminophosphonate and the presence of biomass and dissolved organic matter in a real wastewater treatment plant probably greatly reduce the significance of this reaction. This is a point that should be discussed further or possibly illustrated by carrying out new experiments, for example in the presence of biomass and/or dissolved effluent organic matter. It is well known that organic matter can adsorb onto manganese oxides and strongly inhibit their reactivity.

Response:

We would like to thank the reviewer for the positive feedback regarding the conceptualization of our work. We agree with the reviewer that information on the reaction investigated in

environmentally relevant matrices is essential to strengthen the environmental implications of our findings. Therefore, we included results from additional experiments in a wastewater matrix (sampled after mechanical treatment) in the revised manuscript. The new experimental results are presented in a new subchapter “Formation of glyphosate and AMPA from DTPMP in wastewater matrix” as follows:

Lines 272 to 295:

To address the environmental relevance of the observations in pure water, experiments containing 1 g/L MnO₂ and 1 mM Mn²⁺ were conducted in wastewater (pH 8, sterile filtrated; see Materials & Methods for details on the wastewater sample). In control experiments with unspiked wastewater with and without MnO₂ or Mn²⁺, negligible glyphosate and AMPA concentrations were occasionally detected (see Fig. S6). Dissolved manganese in the wastewater sample was below the detection limit (<0.04 mg/L).

DTPMP transformation kinetics with 1.0 g/L MnO₂ in the wastewater matrix were slower than in MES buffer at pH 6 (see Fig. S7 b) in line with a lower oxidation potential of manganese oxides at higher pH^{47,48}, as well as increased electrostatic repulsion between DTPMP and the mineral surface (point of zero charge of 5.6)^{30,49}. Furthermore, differences between the experiments with MES buffer and wastewater are likely due to the complex wastewater-matrix containing organics (such as other complexing agents) and cations (e.g., calcium), which were reported to influence APP transformation and sorption^{18,50}. Glyphosate and AMPA yields were up to 0.06 mol-% (glyphosate) and 10.3 mol-% (AMPA) after 168 hours. However, in wastewater spiked with 1 mM Mn²⁺, DTPMP transformation kinetics were faster and had higher glyphosate and AMPA yields compared to MES-buffered pure water at pH 6 (see Fig. 5.) In wastewater spiked with 1 mM Mn²⁺, the highest glyphosate yield (0.42 mol-%) of all experiments conducted in this study was observed (see Table 2), albeit only after 240 h. Possibly, the reaction kinetics are faster at pH 8 due to stronger complex formation of Mn^{II} and DTPMP, such as shown for ATMP²⁸. At higher pH, less protons compete with metal ions present in solution and DTPMP is more negatively charged⁵¹. Furthermore, Mn^{III} complexes might play a role at higher pH. While the stability of Mn^{III} complexes varies depending on the ligand, certain ligands including desferrioxamine B show higher stability of Mn^{III} complexes at pH values between 7-11⁵².

Concerning the initial concentration of DTPMP, we agree with the reviewer that an initial DTPMP concentration of 1 mM is unlikely in environmental or technical systems. Nevertheless, as this is a proof-of-concept study, we conducted the experiments with DTPMP concentrations in a concentration range, that allowed us to quantify glyphosate

without involving enrichment steps, as this adds another complexity dimension and source of error. Investigating realistic DTPMP concentrations in WWTP influents – <0.05-2 μM (Nowack & Stone 1998) – would already require enrichment for the parent compound DTPMP using our IC-PAD method. For glyphosate, if assuming similar molar yields as detected now, enrichment by a factor of 200-1000 would be needed for the LC-QQQ method described in the paper. The maximum published enrichment factor for glyphosate we could find was 18, using molecularly imprinted polymers⁴⁵. Thus, if feasible at all, it would first require method development.

Consideration of realistic DTPMP concentrations (0.05 to 2 μM) is highly recommended at later stages in more applied and case specific work.

Below are some detailed comments.

1. The mechanism in Figure 1 needs to be revised slightly. If the initial cleavage takes place on the C-P bond, a methyl group must remain on the nitrogen, so the amine remains a tertiary amine and not a secondary amine as shown. In fact, the initial step would involve cleavage of the C-N bonds at both nitrogen atoms. These concomitant cleavages probably explain the small quantities of glyphosate formed during this transformation, compared with IDMP that is directly produced from one C-N bond cleavage.

Response:

We thank the reviewer for the comment and agree. We modified the scheme accordingly.

Fig. 1 Schematic representation of the formation of phosphate, AMPA, IDMP and glyphosate (proposed) from DTPMP. Phosphate can form via one C-P bond cleavage (iv), IDMP is formed via one C-N bond cleavage (ii), while AMPA is formed via two C-N bond cleavages (i, ii). We propose one pathway for the formation of glyphosate from DTPMP via two C-N bond cleavages (i, iii) and oxidation of the terminal C first to the aldehyde (v) and then to the carboxylic acid (vi). The symmetry of the DTPMP molecule, which contains five phosphonate groups, allows multiple equivalent bond cleavages to lead to the same resultant product. For clarity, only one representative option for each potential cleavage is illustrated. All compounds are depicted in their fully deprotonated forms.

We want to elaborate why we initially marked the C-P bond cleavage (copied from above): We have shown one C-N and one C-P cleavage in the scheme, because PO_4^{3-} is a major and widely known transformation product of DTPMP and all other phosphonates. But – you are right – in order to yield glyphosate, we ultimately need two C-N bond cleavages, as glyphosate contains a secondary amine function. This is why we updated Fig. 1, now showing two C-N bond cleavages but also the formation of phosphate.

2. Proposed protocol also involves the formation of an aldehyde compound as intermediate. This compound can be detected using the FMOC derivatization method. Have any tests been carried out to identify this compound, for example by LC-HRMS?

Response:

We thank the reviewer for this comment. We did not analyse aldehyde formation in this work. However, in our study on the transformation of IDMP on MnO₂ in detail (Röhnelt et al. 2023), we found the corresponding aldehyde (N-formyl-AMPA) formed from IDMP. This provided the necessary grounds for our hypothesis that glyphosate formation from DTPMP is plausible in the first place.

3. Line 81. Meaning of IDMP is missing

Response:

We thank the reviewer and added the abbreviation for IDMP:

Line 70 to 72:

Transformation of APPs (photolysis, Mn²⁺/O₂, MnOOH) with AMPA, iminodi(methylene phosphonate) (IDMP) and phosphate as major TPs (see Fig. 1) is well documented in the literature^{17-19,26-28}.

4. Line 302. Unit for conductivity is μS/cm not μS/cm².

Response:

We thank the reviewer and corrected the mistake accordingly:

Lines 389 to 391:

The water used for the experiments and IC-PAD measurements was purified by an ultrapure water purification system (Barnstead, GenPure Pro, Thermo Scientific, Waltham, MA, USA) down to a conductivity below 0.06 μS/cm.

5. Line 315. I assume it should read “was purchased” instead of “and purchased”.

Response:

We agree with the reviewer. The sentence now reads:

Lines 404 to 407:

For IC-ICP-MS eluent preparation, aqueous ammonia solution (25-27 %, for trace analysis) was purchased from VWR International LLC (Radnor, PA, USA) and diethylenetriaminepentaacetic acid (DTPA) from Honeywell/Fluka (Charlotte, NC, USA).

6. Line 319. Indicate in the protocol how MnO₂ suspensions were mixed during experiments or give reference.

Response:

We thank the reviewer and agree. The text now reads as follows:

Lines 420 to 422:

The reaction suspensions were shaken in an overhead-shaker with a speed of 25 rpm. For a detailed description of the sampling procedure, we refer to Röhnelt *et al.*²⁶.

7. Line 320. Falcon® instead of falcon

Response:

We agree partially with the reviewer. Indeed, we did not use Falcon tubes. Therefore, we use the term “centrifugation tubes” and added the manufacturer in the revised manuscript. It now reads:

Lines 413 to 414:

The experiments (duplicates with one control) were conducted in 50 mL centrifugation tubes (polypropylene, Fisher Scientific, Waltham, MA, USA) in the presence of ambient [...].

8. Line 327. It is stated here that due to the high sorption of DTPMP, the residues were desorbed by adding sodium phosphate and sodium hydroxide. It is not clear what the residues are here, i.e. residual DTPMP or all transformation products, including AMPA and glyphosate. In other words, please state whether the desorption procedure was used prior to any analysis of the transformation products. If so, how does this procedure affect the LOD and LOQ and what is the proportion of sorbed TPs?

Response:

We thank the reviewer for the comment. The residue refers to the mineral pellet after centrifugation, opposed to the supernatant. In order to improve the clarity of the description, we updated the sentence, which now reads:

Lines 422 to 424:

To desorb residual analytes from the mineral pellet after centrifugation and sampling the supernatant (aqueous phase), the mineral pellet was treated with 0.1 M NaOH and 0.1 M NaH₂PO₄ in the ultrasonic bath for 30 minutes⁶⁴.

9. Table 2. Typo: LOQ for glyphosate is 0.198 μ M not 0.189 μ M.

Response:

We thank the reviewer and agree.

The relatively high LOQ value was caused by high background in this measurement sequence, and together with the comparably low glyphosate concentrations in this experiment, we weren't able to quantify glyphosate. This is why we repeated the glyphosate measurement in this experiment after thoroughly cleaning the LC-QQQ. This time, the background was lower, leading to a lower LOQ (0.04 μ M) and thus, we could then quantify glyphosate with a concentration of 0.26 \pm 0.02 μ M at the last timepoint, corresponding to a yield of 0.03 \pm 0.00 mol%.

We added the new experimental results to Table 2 and moved it to the new subchapter.

Additionally, we now thought it would be advisable to add a table with LOD/LOQ values for glyphosate and AMPA formation to the Supporting Information. This can be found as Table S1 in the Supporting Information.

Table S 1: LOD and LOQ values for glyphosate and AMPA in µg/L for each measurement sequence for the LC-QQQ measurements. The approach used to derive those LOD/LOQ values is described in Materials & Methods. The controls are displayed below the respective experiment.

Experiment			glyphosate		AMPA	
Manganese	O ₂	matrix	LOD	LOQ	LOD	LOQ
1.0 g/L MnO ₂	oxic	MES	10.48	17.44	6.41	10.03
0.1 g/L MnO ₂	oxic	MES	5.09	12.13	6.31	10.73
1.0 g/L MnO ₂	anoxic	MES	9.80	33.47	9.49	20.94
0.1 g/L MnO ₂	anoxic	MES	3.57	6.98	9.49	20.94
1 mM Mn ²⁺	oxic	MES	4.63	8.64	4.42	8.94
- Control 1			4.63	8.64	4.42	8.94
1.0 g/L MnO ₂	oxic	wastewater	4.89	10.14	4.42	8.94
- Control 1			3.57	6.98	1.71	4.83
- Control 2			5.50	11.00	5.36	12.04
- Control 3			3.57	6.98	1.71	4.83
1 mM Mn ²⁺	oxic	wastewater	4.63	8.64	5.36	12.04
- Control 1			4.63	8.64	4.42	8.94

References

1. Post, J. E. Manganese oxide minerals: Crystal structures and economic and environmental significance. *Proc. Natl. Acad. Sci. USA* **96**, 3447–3454 (1999).
2. Andrea Roskosch & Patric Heidecke. *Klärschlamm Entsorgung in Der Bundesrepublik Deutschland*.
https://www.umweltbundesamt.de/sites/default/files/medien/376/publikationen/2018_10_08_uba_fb_klaerschlamm_bf_low.pdf (2018).
3. Milik, J., Pasela, R., Lachowicz, M. & Chalamoński, M. The concentration of trace elements in sewage sludge from wastewater treatment plant in Gniewino. *Journal of Ecological Engineering* **18**, 118–124 (2017).
4. Krauskopf, K. B. Geochemistry of micronutrients. in *Micronutrients in Agriculture* (eds. Mortvedt, J. J., Cox, F. R., Shuman, L. M. & Welch, R. M.) 7–40 (Soil Science Society of America, Madison, WI, USA, 1972).
5. Fuller, W. H. & Warwick, A. W. *Soils in Waste Treatment and Utilization: Volume I: Land Treatment*. vol. 1 (CRC Press, Boca Raton, FL, USA, 1985).
6. Ghosh, S. K. Diversity in the Family of Manganese Oxides at the Nanoscale: From Fundamentals to Applications. *ACS Omega* vol. 5 25493–25504 Preprint at <https://doi.org/10.1021/acsomega.0c03455> (2020).
7. Geszvain, K. *et al.* The molecular biogeochemistry of manganese(II) oxidation. in *Biochemical Society Transactions* vol. 40 1244–1248 (2012).
8. Nowack, B. & Stone, A. T. Homogeneous and heterogeneous oxidation of nitrilotrismethylenephosphonic acid (NTMP) in the presence of manganese(II, III) and molecular oxygen. *Journal of Physical Chemistry B* **106**, 6227–6233 (2002).
9. Nowack, B. & Stone, A. T. Manganese-catalyzed degradation of phosphonic acids. *Environ Chem Lett* **1**, 24–31 (2003).
10. Martin, P. R., Buchner, D., Jochmann, M. A., Elsner, M. & Haderlein, S. B. Two Pathways Compete in the Mn(II)-Catalyzed Oxidation of Aminotrimethylene Phosphonate (ATMP). *Environ Sci Technol* **56**, 4091–4100 (2022).
11. Drzyzga, D., Forlani, G., Vermander, J., Kafarski, P. & Lipok, J. Biodegradation of the aminopolyphosphonate DTPMP by the cyanobacterium *Anabaena variabilis* proceeds via a C–P lyase-independent pathway. *Environ Microbiol* **19**, 1065–1076 (2017).

12. Schowanek, D. & Verstraete, W. Phosphonate Utilization by Bacterial Cultures and Enrichments from Environmental Samples. *Appl Environ Microbiol* **56**, 895–903 (1990).
13. Riedel, R. *et al.* Laundry Isolate Delftia sp. UBM14 Capable of Biodegrading Industrially Relevant Aminophosphonates. *Microorganisms* **12**, (2024).
14. Riedel, R. *et al.* Novel standard biodegradation test for synthetic phosphonates. *J Microbiol Methods* **212**, (2023).
15. Riedel, R. *et al.* Biodegradation of selected aminophosphonates by the bacterial isolate Ochrobactrum sp. BTU1. *Microbiol Res* **280**, (2024).
16. Röhnelt, A. M., Martin, P. R., Buchner, D. & Haderlein, S. B. Transformation of Iminodi(methylene phosphonate) on Manganese Dioxides - Passivation of the Mineral Surface by (Formed) Mn²⁺. *Environ Sci Technol* **57**, 11958–11966 (2023).
17. Remucal, C. K. & Ginder-Vogel, M. A critical review of the reactivity of manganese oxides with organic contaminants. *Environmental Sciences: Processes and Impacts* **16**, 1247–1266 (2014).
18. Lyu, Y. *et al.* Catalytic oxidation of toluene over MnO₂ catalysts with different Mn (II) precursors and the study of reaction pathway. *Fuel* **262**, 116610 (2020).
19. Bouchaala, L., Charchar, N., Sahraoui, H. & Gherib, A. Assessment of wastewater biological treatment efficiency and mapping of WWTPs and LTPs in Algeria. *J Environ Health Sci Eng* **19**, 1153–1169 (2021).
20. Kuhn, R. *et al.* The influence of selected bivalent metal ions on the photolysis of diethylenetriamine penta(methylenephosphonic acid). *Chemosphere* **210**, 726–733 (2018).
21. Grandcoin, A., Piel, S. & Baurès, E. AminoMethylPhosphonic acid (AMPA) in natural waters: Its sources, behavior and environmental fate. *Water Research* vol. 117 187–197 Preprint at <https://doi.org/10.1016/j.watres.2017.03.055> (2017).
22. Lesueur, C., Pfeffer, M. & Fuerhacker, M. Photodegradation of phosphonates in water. *Chemosphere* **59**, (2005).
23. Jaworska, J., Van Genderen-Takken, H., Hanstveit, A., Van De Plassche, E. & Feijtel, T. Environmental risk assessment of phosphonates, used in domestic laundry and cleaning agents in the Netherlands. *Chemosphere* 655–665 (2002).

24. Popov, K., Rönkkömäki, H. & J Lajunen, L. H. *Critical Evaluation of Stability Constants of Phosphonic Acids (IUPAC Technical Report)*. P.R. China); National Representatives: R. Apak (Turkey vol. 73 (2001).
25. Nowack, B. & Stone, A. T. The influence of metal ions on the adsorption of phosphonates onto goethite. *Environ Sci Technol* **33**, 3627–3633 (1999).
26. Gledhill, W. E. & Feijtel, T. C. J. Environmental Properties and Safety Assessment of Organic Phosphonates Used for Detergent and Water Treatment Applications. in *The Handbook of Environmental Chemistry – Detergents* 261–285 (1992). doi:10.1007/978-3-540-47108-0_8.
27. Ruffolo, F. et al. The Microbial Degradation of Natural and Anthropogenic Phosphonates. *Molecules* vol. 28 Preprint at <https://doi.org/10.3390/molecules28196863> (2023).
28. Schwientek, M. et al. Glyphosate contamination in European rivers not from herbicide application? *Water Res* **263**, 122140 (2024).
29. European Phosphonates Association. *Input to the Revision of the EU Ecolabels Related to Detergents*. <https://www.phosphonates.org/images/Images/Documents/EPA%20phosphonate%20input%20detergent%20Ecolabel.pdf> (2015).
30. Nowack, B. Environmental chemistry of phosphonates. *Water Res* **37**, 2533–2546 (2003).
31. Hansel, C. M. & Learman, D. R. Geomicrobiology of Manganese. in *Ehrlich's Geomicrobiology* (eds. Ehrlich, H. L., Newman, D. K. & Kappler, A.) 401–452 (CRC Press, Taylor & Francis Group, Boca Raton, FL, USA, 2016).
32. Tebo, B. M. et al. Biogenic manganese oxides: Properties and mechanisms of formation. *Annual Review of Earth and Planetary Sciences* vol. 32 287–328 Preprint at <https://doi.org/10.1146/annurev.earth.32.101802.120213> (2004).
33. Nowack, B. & Stone, A. T. Degradation of nitrilotris(methylenephosphonic acid) and related (amino)phosphonate chelating agents in the presence of manganese and molecular oxygen. *Environ Sci Technol* **34**, 4759–4765 (2000).
34. Barrett, K. A. & McBride, M. B. Oxidative degradation of glyphosate and aminomethylphosphonate by manganese oxide. *Environ Sci Technol* **39**, 9223–9228 (2005).

35. Li, H. & Jaisi, D. P. Competition of Sorption and Degradation Reactions during Glyphosate Degradation by Ferrihydrite/ δ -Manganese Oxide Composites. *ACS Earth Space Chem* **3**, 1362–1370 (2019).
36. Li, H., Joshi, S. R. & Jaisi, D. P. Degradation and Isotope Source Tracking of Glyphosate and Aminomethylphosphonic Acid. *J Agric Food Chem* **64**, 529–538 (2016).
37. Li, H., Wallace, A. F., Sun, M., Reardon, P. & Jaisi, D. P. Degradation of Glyphosate by Mn-Oxide May Bypass Sarcosine and Form Glycine Directly after C-N Bond Cleavage. *Environ Sci Technol* **52**, 1109–1117 (2018).
38. Benbrook, C. M. How did the US EPA and IARC reach diametrically opposed conclusions on the genotoxicity of glyphosate-based herbicides? *Environ Sci Eur* **31**, (2019).
39. Tarazona, J. V. *et al.* Glyphosate toxicity and carcinogenicity: a review of the scientific basis of the European Union assessment and its differences with IARC. *Archives of Toxicology* vol. 91 2723–2743 Preprint at <https://doi.org/10.1007/s00204-017-1962-5> (2017).
40. Caiati, C., Pollice, P., Favale, S. & Lepera, M. E. The Herbicide Glyphosate and Its Apparently Controversial Effect on Human Health: An Updated Clinical Perspective. *Endocr Metab Immune Disord Drug Targets* **20**, 489–505 (2019).
41. Myers, J. P. *et al.* Concerns over use of glyphosate-based herbicides and risks associated with exposures: A consensus statement. *Environmental Health* **15**, (2016).
42. Venditti, S., Kiesch, A. & Hansen, J. Fate of glyphosate and its metabolite AminoMethylPhosponic acid (AMPA) from point source through wastewater sludge and advanced treatment. *Chemosphere* **340**, (2023).
43. Armbruster, D., Rott, E., Minke, R. & Happel, O. Trace-level determination of phosphonates in liquid and solid phase of wastewater and environmental samples by IC-ESI-MS/MS. *Anal Bioanal Chem* **412**, 4807–4825 (2019).
44. Rott, E., Happel, O., Armbruster, D. & Minke, R. Behavior of PBTC, HEDP, and aminophosphonates in the process of wastewater treatment. *Water (Switzerland)* **12**, (2020).
45. Surapong, N. & Burakham, R. Magnetic Molecularly Imprinted Polymer for the Selective Enrichment of Glyphosate, Glufosinate, and Aminomethylphosphonic Acid

Prior to High-Performance Liquid Chromatography. *ACS Omega* **6**, 27007–27016 (2021).

REVIEWERS' COMMENTS

Reviewer #1 (Remarks to the Author):

The authors have done a great job answering my concerns/questions and revising the manuscript. Though my concerns on reduction and transformation of Mn-minerals are not resolved in this revision, I agree that these analyses are not the main focus of this study and can be done in future studies. Overall, the manuscript is well improved.

We thank the reviewer for that positive criticism and are happy that we could improve the manuscript.

Reviewer #2 (Remarks to the Author):

I appreciate the efforts that the authors have made to address comments made by the three reviewers and am happy to see that they have included some additional experiments to address several of my main concerns about the extrapolation of the data in laboratory experiments to conditions in the environment.

I recognize that the intention of this manuscript is to demonstrate a proof of concept that MnO₂ can oxidize DTPMP under conditions encountered in sewage but I am still concerned about the issue of the MnO₂ polymorphs and their concentration. If the editor is comfortable with the publication of a result that is still somewhat speculative I would not object, as long as the authors are more direct about the nature of the extrapolation that they have made. With respect to the concentration of Mn-oxides present in the samples, I accept the idea that wastewater solids (e.g., sludge or mixed liquor suspended solids) may contain 1 g/kg of Mn and that these experiments may approximate the concentrations detected in wet wastewater sludge (as the authors indicate in their response to my comment but this is not the question that the authors have set out to answer nor is it the meaningful question for the research. Sewage consists of a small amount of wastewater solids suspended in a relatively large volume of water. For example, if the sewage contained 1 gram of solids per liter of water then these experiments would have been conducted with Mn concentrations that are three orders of magnitude higher than those that are encountered in the environment. Perhaps biogenic Mn-oxides are much more reactive than the particles used in these experiments, but a better acknowledgment of this limitation is needed.

We sincerely thank the reviewer for once again emphasizing the importance of the environmental relevance of our study. However, as the reviewer acknowledges in the first sentence of their comment, our study was designed as a proof-of-concept investigation. This inherently means that achieving environmentally relevant concentrations—whether for DTPMP or MnO₂—was not our primary objective. Instead, our focus was to demonstrate that the chemical reaction is indeed feasible under controlled laboratory conditions. We respectfully argue that our results are not speculative, as we clearly demonstrated glyphosate formation in all conducted experiments covering a relevant range of environmental conditions. Throughout the manuscript, and particularly in the last paragraphs, we have emphasized that our findings are based on laboratory experiments and that further research is required to explore how DTPMP behaves under real-world environmental conditions. Specifically, we state:

Lines 341 to 345:

“While we could demonstrate that the reaction is chemically feasible in the laboratory, future research should elucidate in detail how environmental conditions affect glyphosate formation from DTPMP and related APPs, the formation and identification of key intermediates and field studies including yields in wastewater treatment plants.”

In addition to the proof-of-concept argument, we would like to address the reviewer’s concern regarding Mn concentrations being orders of magnitude too high due to sludge being dispersed in water. We believe this argument may not be entirely applicable, as APPs such as DTPMP are known to strongly adsorb to sediments and suspended matter. As such, they are primarily present in the solid phase of suspensions rather than freely dissolved in water. This behavior is well-documented by Rott et al. (2020)¹, who investigated the distribution of polyphosphonates across sediment, suspended matter, and aqueous phases in wastewater effluents and river water. Considering this, it can be expected that in wastewater treatment plants, the Mn concentration per volume of water plays a minor role compared to factors such as the Mn content of sludge particles, their specific surface area, and the surface concentration of Mn on these particles (as MnO₂ often exists as coatings). Nonetheless, these considerations are beyond the scope of our study, as our primary goal was to demonstrate the feasibility of glyphosate formation via Mn-driven oxidation of DTPMP under controlled laboratory conditions.

Similarly, I still find it surprising that the authors have been so casual about the form of Mn-oxides used in the experiments. It is hardly surprising that some random Mn oxide on the shelf of the lab included a few different crystalline phases and some amorphous oxides when analyzed by XRD. The presence of amorphous phases does not convince me that this is a surrogate for biogenic Mn-oxides. The absence of any data on where the Mn oxide originated (if it is from a mineral collection) or how it was synthesized (if it is synthetic) is a minimum expectation for a scientific study. As it stands, the experiments reported here are irreproducible because the type of Mn-oxide remains undefined. If the editor is willing to publish this work without such information a clearer statement is needed indicating that only one type of Mn oxide was studied, that its provenance is uncertain and that Mn oxides in sewage might behave in a much different manner because they will form by different mechanisms and will be more intimately associated with organic matter is necessary.

We appreciate the reviewer’s feedback and would like to clarify any potential misunderstanding. The Mn oxide’s origin is explicitly stated in the “Methods” section (purchased from Carl Roth (Karlsruhe, Germany)). The use of commercially available MnO₂ ensures reproducibility, as this material can be obtained and verified by other researchers. To further address the reviewer’s concern regarding reproducibility, we are happy to provide additional details about the exact batch number of the MnO₂ used, which allows access to the manufacturer’s certificate of analysis:

Lines 362 to 364:

“All chemicals were purchased from Merck (Darmstadt, Germany) in the highest available purity, if not described differently. The manganese dioxide (Manganese^{IV}oxide, ≥98%; MnO₂, Batch No. 168267405) was purchased from Carl Roth (Karlsruhe, Germany).”

Furthermore, we now provide additional characterization data for the MnO₂, such as extractable Mn(III) content or TXRF results on intercalated mono- and bivalent cations. This information is now included the Supporting Information:

“Further mineral analysis

The extractable Mn^{III}-content (quantified by pyrophosphate extraction and UV/vis spectroscopy) was 1 % of total manganese. With regard to the surface area, the extractable Mn^{III}-content was 1.76 nmol/m². Further, total reflection X-ray fluorescence (TXRF) revealed just minor amounts of mono- and bivalent cations (K: <0.001, Ca and Ba: ≤0.003 molar ratio vs Mn), which validates the low Mn^{III}-content.”

The fact that glyphosate does not form when DTPMP is added to raw wastewater (without amending with Mn-oxides) is evidence that there is an issue that requires further study.

We appreciate the reviewer's attention to detail. However, the stability of DTPMP in the raw wastewater sample can be explained by the experimental conditions. The wastewater was sterile filtered, removing all particles >0.22 μm, including sludge and suspended matter. As we hypothesize that the reaction with MnO₂ is likely responsible for glyphosate formation from DTPMP in WWTPs, this transformation would primarily occur in the solid phase – specifically in sludge or suspended matter, which were eliminated in our filtered sample.

Furthermore, the Mn content of the wastewater sample was below the detection limit of 0.04 mg/L Mn. Given these conditions, the stability of DTPMP in the wastewater sample is consistent with our hypothesis. This observation underscores the potential significance of sludge particles and/or Mn in facilitating DTPMP transformation and subsequent glyphosate formation.

Reviewer #3 (Remarks to the Author):

This manuscript presents the results of a study showing that the transformation of EDTMP phosphonate by MnO₂ can be a source of glyphosate, a pesticide widely used in agriculture for weed control. The authors have made significant changes to the original version. The objectives of the study are clearer and the new experiments with wastewater reinforce its relevance (but certain characteristics of wastewater are not usual, see below). The authors have responded very carefully to all the reviewers' comments and made the appropriate corrections.

We want to thank the reviewer for the positive feedback.

However, there are still minor issues that need to be addressed before possible publication.

Minor detailed comments:

- Line 232. Yield of glyphosate is 0.03 mol% (corresponding to 0.3 μM) not 0.03 μM.

Yes, that is correct, we changed it accordingly.

- Line 238. Please rephrase “Glyphosate and AMPA concentrations only rose after 1.5 respectively 1 hour, even though DTPMP” by, for example, “Glyphosate and AMPA concentrations only rose after 1.5 and 1 hour, respectively, even though DTPMP...”

We thank the reviewer for that suggestion and changed it accordingly.

- Line 240. I would change “The maximum AMPA and glyphosate concentrations formed differed between the four experiments » by “The maximum concentrations of AMPA and glyphosate formed differed between the four experiments”

We thank the reviewer for that suggestion and changed it accordingly.

- Line 285. In “Glyphosate and AMPA yields were up to 0.06 mol-% (glyphosate) and 10.3 mol-% (AMPA) after 168 hours”, I would specify that it is in wastewater.

We thank the reviewer for that suggestion and changed it accordingly.

- Table 8. The effluent conductivity, ~100 mS/cm (i.e. 66 g NaCl/L equivalent), is particularly high for wastewater and given the ion concentrations in Table 7; this conductivity is probably not correct. Similarly, DOC (11.6 mg/L) seems relatively low for a wastewater influent (usual values for DOC are more between 100 and 300 mg/L in influent) and comparatively to the COD of 195 mg/L (which is low for influent but high for a treated wastewater). Is this sample characteristic of a wastewater influent? Is this sample a wastewater influent or a treated wastewater effluent? Please clarify.

We thank the reviewer for noticing those incongruities. Indeed, after reconfirming with the WWTP the actual value for wastewater conductivity on that day (24 h mix sample) was 691 μ S/cm. The value is rather low, as it rained heavily the previous day and night. On an average dry day, the wastewater volume treated in this treatment plant usually amounts to 25 000 to 30 0000 m³/day. On September 9, 2024, the total volume was roughly 80 000 m³/d. Hence, the wastewater was diluted on that day. This explains the relatively low DOC and COD values. On an average dry day, the COD is around 300 mg/L and the conductivity around 1.2 mS/cm.

The wastewater used for the transformation experiments was sampled after the screen and grit chamber, but before the primary settling tank. This indicates that the wastewater had only undergone preliminary treatment. The sampling point for monitoring of the wastewater within the WWTP (values are indicated for the “24 h mix sample” in the manuscript, Table S7) instead is after the screen but before the grit chamber, hence it underwent even less treatment.

To account for this comment, we amended the description of the wastewater sample as follows:

Lines 434 to 436:

“The wastewater was sampled at 10 am on September 9, 2024, from the municipal wastewater treatment plant in Lustnau (Tübingen, SW Germany), after the screen and grit chamber, but before the primary settling tank.”

Lines 562 to 571:

“The wastewater sample taken from the WWTP in Lustnau (Tübingen, Germany) on September 9, 2024, had a pH value of 7.94. [...] Supplementary Table S7 contains additional information on a 24h-mixed wastewater sample monitored by the WWTP Lustnau. The sampling point for the latter lies after the screen but before the grit chamber. Due to heavy rainfall the day and night before wastewater sampling the wastewater volume treated in the WWTP in Lustnau (Tübingen, Germany) on September 9, 2024, amounted to ~80 000 m³/d compared to 25 000 – 30 000 m³/d on an average dry day.”

- Supporting information

Caption of Figure S-5. The following sentence is not clear. Please rephrase. “Aqueous concentration of DTPMP quantified by means of IC-ICP-MS and total concentrations of glyphosate and AMPA quantified by means of LC-QQQ after derivatization in the replica the experiments over 96 hours using 1.0 g/L MnO₂ under oxidic conditions.”

We thank the reviewer for noticing and rephrased the caption of Figure S5 accordingly:

“Aqueous concentrations of DTPMP (black) quantified by means of IC-ICP-MS and total concentrations of glyphosate (red) and AMPA (blue) quantified by means of LC-QQQ in the longtime replica of the experiment using 1.0 g/L MnO₂ in ultrapure buffered water (pH 6) under oxidic conditions over 96 hours. Error bars represent absolute errors between experimental duplicates.”

References

1. Rott, E., Happel, O., Armbruster, D. & Minke, R. Influence of wastewater discharge on the occurrence of PBTC, HEDP, and aminophosphonates in sediment, suspended matter, and the aqueous phase of rivers. *Water*, **12**, 1–23 (2020).